# Asymmetry Learning for Counterfactual-Invariant Classification in OOD Tasks

**S Chandra Mouli**
Department of Computer Science
Purdue University
chandr@purdue.edu

**Bruno Ribeiro**
Department of Computer Science
Purdue University
ribeiro@cs.purdue.edu

## Abstract

Generalizing from observed to new related environments (out-of-distribution) is central to the reliability of classifiers. However, most classifiers fail to predict label $Y$ from input $X$ when the change in environment is due a (stochastic) input transformation $T^{\text{te}} \circ X'$ not observed in training, as in training we observe $T^{\text{tr}} \circ X'$, where $X'$ is a hidden variable. This work argues that when the transformations in train $T^{\text{tr}}$ and test $T^{\text{te}}$ are (arbitrary) symmetry transformations induced by a collection of known $m$ equivalence relations, the task of finding a robust OOD classifier can be defined as finding the simplest causal model that defines a causal connection between the target labels and the symmetry transformations that are associated with label changes. We then propose a new learning paradigm, *asymmetry learning*, that identifies which symmetries the classifier must break in order to correctly predict $Y$ in both train and test. *Asymmetry learning* performs a causal model search that, under certain identifiability conditions, finds classifiers that perform equally well in-distribution and out-of-distribution. Finally, we show how to learn counterfactually-invariant representations with *asymmetry learning* in two simulated physics tasks and six image classification tasks.

## 1 Introduction

A significant challenge in classification tasks happens when the test distribution differs from the training distribution (i.e., the task requires out-of-distribution (OOD) generalization), since not accounting for the distribution shift can lead to poor generalization accuracy (Geirhos et al., 2020; Hu et al., 2020; Koh et al., 2020; D'Amour et al., 2020). If the learner sees examples from the test distribution, finding a classifier invariant to the distribution shift can still be a data-driven task (e.g., classical domain adaptation Ben-David et al. (2007); Muandet et al. (2013); Zhao et al. (2019)). This includes cases such as invariant risk minimization (Arjovsky et al., 2019) and its generalizations (Bellot & van der Schaar, 2020), where the training data and the test data distributions overlap in a way that can be exploited by data-driven algorithms (Creager et al., 2021; Krueger et al., 2021; Rosenfeld et al., 2020).

However, if the learner sees no examples from the test distribution, the task is not purely data-driven and requires assumptions about the data generation process. More formally, our work considers general OOD tasks with training distribution $P(Y^{\text{tr}}, X^{\text{tr}})$, where $X^{\text{tr}} := T^{\text{tr}} \circ X^{\dagger}$, with $X^{\dagger}$ as a hidden variable with distribution $P(X^{\dagger})$ and $T^{\text{tr}} \in \mathcal{T}$ is a random input transformation in training $T^{\text{tr}} : \mathcal{X} \to \mathcal{X}$, where $t \circ x$ is the application of transformation $t \in \mathcal{T}$ on $x \in \mathcal{X}$. The difference between train and test is a change in input transformation with $Y^{\text{te}} := Y^{\text{tr}}$ and $X^{\text{te}} := T^{\text{te}} \circ X^{\dagger}$, where $P(T^{\text{tr}}) \neq P(T^{\text{te}})$. We are interested in learning an invariant classifier that *generalizes well* in held out examples from the training and test distributions.

The definition of transformation matters in this task. We first seek to generalize the existing literature on transformation invariances, e.g. (Shawe-Taylor, 1993; Kondor & Trivedi, 2018; Finzi et al., 2021; Maron et al., 2018; Murphy et al., 2019b; Mouli & Ribeiro, 2021; Bronstein et al., 2017). Our transformations are tied to equivalence relations rather than transformation groups, which frees them from the need to have inverses (in order to form a transformation group). Our transformations may not have inverses.

We also explain why the task of learning an invariant OOD classifier is not, in general, solvable via traditional data augmentation. Before we continue describing our OOD learning task, it is important to clarify the connection between Pearl's causal hierarchy and invariant representation learning.

**Pearl's causal hierarchy and invariant representation learning.** Pearl's causal hierarchy (Pearl & Mackenzie, 2018; Bareinboim et al., 2020)) has three layers: Observational (Layer 1), interventional (Layer 2), and counterfactual (Layer 3). Upper layers can perform lower layer tasks, but not vice-versa (see Bareinboim et al. (2020)). *Tasks should be described using the lowest layer that can solve them.*

*Layer 1:* Any task that can be performed without constraints on the causal model, i.e., by data alone, is observational (Layer 1). Traditional domain adaptation is a Layer 1 task. Note that a classifier that performs well OOD is itself a Layer 1 classifier, since it tries to predict $P(Y^{\text{te}}|X^{\text{te}})$.

*Layer 2:* Without observations from $P(X^{\text{te}})$ and/or $P(Y^{\text{te}}|X^{\text{te}})$, *learning an OOD classifier* requires some assumptions about the data generation process (Layers 2 or 3 assumptions). Data augmentation is traditionally an interventional task (Layer 2), with new interesting methods increasingly using causal language (Ilse et al., 2021; Teney et al., 2020). For instance, in a task predicting an image's foreground, knowing how to act on an image in training $X^{\text{tr}}$ to change the background seen in training to the backgrounds seen in test $X^{\text{te}} = T \circ X^{\text{tr}}$ with a transformation $T$, implies we know how to predict $P(Y|X, do(T))$.

*Layer 3:* Counterfactuals are the most challenging task. We start our description with an example. Consider a random continuous transformation $T_2^{\text{tr}}$ (in training) which changes to random transformation $T_2^{\text{te}}$ (in test). Let $X^\dagger$ describe a hidden variable such that $X^{\text{tr}} := T_1 \circ T_2^{\text{tr}} \circ T_3 \circ X^\dagger$ and $X^{\text{te}} := T_1 \circ T_2^{\text{te}} \circ T_3 \circ X^\dagger$, where $T_1$ and $T_3$ are independent continuous random transformations and $P(T_2^{\text{tr}}) \neq P(T_2^{\text{te}})$. Assume the target variable $Y$ depends only on $X^\dagger$, $T_1$, and $T_3$. To counterfactually ask what would have happened to the observed input $x$ if we had forced $do(T_2^{\text{tr}} = \tilde{t}_2)$, we are inquiring about $X(T_2^{\text{tr}} = \tilde{t}_2)|X^{\text{tr}} = x$. Note that $do(T_2^{\text{tr}} = \tilde{t}_2)$ does not change $Y$. Also note that the knowledge of $X^{\text{tr}} = x$ is an indirect statement about $T_2^{\text{tr}}$ since $P(T_2^{\text{tr}}|X^{\text{tr}} = x) \neq P(T_2^{\text{tr}})$. That is, for $x, x' \in \mathcal{X}$,

$$P(X(T_2^{\text{tr}} = \tilde{t}_2) = x'|X^{\text{tr}} = x) = \int_t P(X(T_2^{\text{tr}} = \tilde{t}_2) = x'|T_2^{\text{tr}} = t, X^{\text{tr}} = x)dP(T_2^{\text{tr}} = t|X^{\text{tr}} = x).$$

(1)

Equation (1) and the difference between the causal hierarchy layers will be relevant for our results.

**Contributions.** Our contributions can be described as follows:

1. We introduce a generalization of transformation groups via symmetry transformations tied to equivalence classes that removes the requirement of invertible transformations common in definitions using transformation groups.

2. We introduce the concept of *counterfactual invariant representations for symmetry transformations* and show how it can be described as a counterfactual task for causal structure discovery.

3. Finally, we introduce *asymmetry learning*, which describes a representation regularization that, under a set of assumptions, learns the correct counterfactual invariant OOD classifier.

## 2 SYMMETRIES AND TRANSFORMATIONS

Geometrically, an object is called symmetric if there is a transformation on the object that does not change its shape (in some definition of shape). For example, a square is symmetric with respect to rotations. The notion of symmetry however is not restricted to geometric notions. In general, we can define a mathematical object as symmetric if there is a transformation on the object that returns another object *equivalent* to the first (Rosen, 2008, Chapter 10). It is clear from this definition of symmetry that we first need to define what we mean by equivalent objects. For instance, we say two geometrical objects are equivalent if they have the same shape, but we need a more general definition.

We define an input *symmetry* in a space $\mathcal{X}$ with at least two elements as an equivalence relation $\sim$. An equivalence relation in $\mathcal{X}$ is a binary relation $\sim$ such that for all $a, b, c \in \mathcal{X}$, we have **(i)** $a \sim a$, **(ii)** $a \sim b \iff b \sim a$, and **(iii)** $(a \sim b$ and $b \sim c) \implies a \sim c$. Equivalence relations allow us to

define equivalent objects in $\mathcal{X}$: $\boldsymbol{a} \sim \boldsymbol{b}$ means $\boldsymbol{a}$ is equivalent to $\boldsymbol{b}$. The set of all objects equivalent to some $\boldsymbol{a} \in \mathcal{X}$ is called the equivalence class of $\boldsymbol{a}$, defined as $[\boldsymbol{a}] := \{\boldsymbol{x} \in \mathcal{X} \ : \ \boldsymbol{x} \sim \boldsymbol{a}\}$. Note that one can define $m \geqslant 2$ equivalence relations on the same input space. The equivalence class of $\boldsymbol{x}$ with respect to equivalence relation $k$ is denoted $[\boldsymbol{x}]^{(k)}$, $k = 1, \ldots, m$. Two inputs $\boldsymbol{a}, \boldsymbol{b} \in \mathcal{X}$ might be equivalent under one equivalence relation $\sim_1$, but not equivalent under a different equivalence relation $\sim_2$, that is, we can have both $\boldsymbol{b} \in [\boldsymbol{a}]^{(1)}$ and $\boldsymbol{b} \notin [\boldsymbol{a}]^{(2)}$. Still, even in this last case it is possible that $\boldsymbol{a}$ is equivalent to some other input $\boldsymbol{c} \neq \boldsymbol{b}$ in both equivalence relations, i.e., it is possible $\exists \boldsymbol{c} \in \mathcal{X}, \boldsymbol{c} \neq \boldsymbol{a}$, s.t. $\boldsymbol{c} \in [\boldsymbol{a}]^{(1)} \cap [\boldsymbol{a}]^{(2)}$. We denote the collection of equivalence classes of $\mathcal{X}$ under the equivalence relation $\sim_k$ as the quotient space $\mathcal{X}/\sim_k := \{[x]^{(k)} \mid x \in \mathcal{X}\}$.

*Transformation group example.* Consider the bijective transformations $t : \mathcal{X} \to \mathcal{X}$ of a transformation group $G$, $t \in G$. We now define an equivalence relation over $G$ as $t \circ \boldsymbol{x} \sim_G \boldsymbol{x}$ for all $t \in G$. The equivalence class $[\boldsymbol{x}]^{(G)}$ is $\boldsymbol{x}$'s *orbit* defined as $[\boldsymbol{x}]^{(G)} := \{\boldsymbol{x}' : \exists t \in G, \boldsymbol{x}' = t \circ \boldsymbol{x}\}$. For example, if $G$ is the group that permutes the elements of vectors in $\mathbb{R}^3$, then $(1, 2, 3) \sim_G (2, 1, 3)$.

*Property functions example.* Another way of deriving an equivalence relation is via functions of the input space $z : \mathcal{X} \to \mathbb{R}^p$, where the output $z(\boldsymbol{x})$ is a particular property of the vector $\boldsymbol{x} \in \mathcal{X}$. For example, given an observation of length $T$ from a dynamical system, $\boldsymbol{x} \in \mathbb{R}^{d \times T}$, a possible property function could be $z_{\text{energy}}(\cdot)$ that computes the energy of the dynamical system. Assuming there are $m$ known properties $z_1, \ldots, z_m$ with $z_i : \mathcal{X} \to \mathbb{R}^{p_i}$, we can construct corresponding equivalence relations $\sim_1, \ldots, \sim_m$ such that for any $\boldsymbol{x}, \boldsymbol{x}' \in \mathcal{X}$, $\boldsymbol{x} \sim_i \boldsymbol{x}'$ if $z_j(\boldsymbol{x}) = z_j(\boldsymbol{x}'), \forall j \neq i$. In words, two inputs are equivalent under $\sim_i$ if they have the same properties for all $z_j, j \neq i$.

*Symmetry transformations.* As seen above, symmetries can be defined without defining how the input is transformed to create the equivalence classes, although defining a set of transformations is useful when describing the equivalence class. Given an equivalence relation $\sim$, we can define a set of transformations $\mathcal{T}$ that respect the equivalence relation such that $\forall t \in \mathcal{T}, \forall \boldsymbol{x} \in \mathcal{X}, t \circ \boldsymbol{x} \sim \boldsymbol{x}$. We call $\mathcal{T}$ the set of *symmetry transformations* of $\sim$. Similar to transformations groups, $\mathcal{T}$ always has the identity transformation $t_{\text{id}} \circ \boldsymbol{x} = \boldsymbol{x}$, but in contrast, all the transformations in $\mathcal{T}$ need not be bijective.

*Join of equivalence relations.* Similar to how two groups can be joined to form a larger group, two equivalence relations can be joined to form a coarser equivalence relation. Given two equivalence relations, $\sim_1$ and $\sim_2$, their join $\sim_1 \vee \sim_2$ is defined as: for all $\boldsymbol{x}, \boldsymbol{x}', \boldsymbol{x}(\sim_1 \vee \sim_2)\boldsymbol{x}'$ if and only if there exists a chain of equivalence relations $\boldsymbol{x} \sim_{k_1} \boldsymbol{x}_1, \ \ldots, \ \boldsymbol{x}_{h-1} \sim_{k_h} \boldsymbol{x}'$ with all $k_j \in \{1, 2\}$. It is easy to check that $\sim_1 \vee \sim_2$ is an equivalence relation.

We are now ready to define a general causal model that defines the training and test distributions in our setting.

## 3 SCM FOR SYMMETRY-BASED OOD TASKS

Let $\mathcal{X}, \mathcal{Y}$ denote the input and output spaces respectively. We define our general structural causal model (SCM) as follows. We define $X^\dagger \in \mathcal{X}$ as the unobserved canonically ordered input

$$X^\dagger := g(U_u) , \tag{2}$$

with $U_u$ a background random variable and $g : \mathcal{U}_u \to \mathcal{X}$ is a measurable map. This definition is general enough to define any task.

There are $m$ possible symmetries given in the form of equivalence relations $\sim_1, \ldots, \sim_m$ over the input space $\mathcal{X}$. Let $\mathcal{T}^{(k)}$ denote a set of symmetric transformations $t$ on $\mathcal{X}$ corresponding to the equivalence relation $\sim_k, 1 \leqslant k \leqslant m$. In other words, for all $\boldsymbol{x} \in \mathcal{X}$ and $t \in \mathcal{T}^{(k)}$, we have $(t \circ \boldsymbol{x}) \sim_k \boldsymbol{x}$. Similarly, let $\mathcal{T}$ be the set of all symmetric transformations with respect to the join equivalence relation $\sim_{1,\ldots,m} \equiv \sim_1 \vee \ldots \vee \sim_m$. We can think of transformation $t \in \mathcal{T}$ as a path $\boldsymbol{x} \xrightarrow{t^{(k_1)}} \boldsymbol{x}_1 \cdots \boldsymbol{x}_{h-1} \xrightarrow{t^{(k_h)}} \boldsymbol{x}_h$ that starts at $\boldsymbol{x}$, applies a transformation $t^{(k_1)} \in \mathcal{T}^{(k_1)}$ to get $\boldsymbol{x}_1 \in [\boldsymbol{x}]^{(k_1)}$, and so on until it stops and outputs a value $\boldsymbol{x}_h, h \geqslant 1$.

Let $U_1, \ldots, U_m$ be independent background variables associated with the $m$ symmetries, where $U_i \in \mathcal{U}_i, i = 1, \ldots, m$. These background variables together select a function $t(U_1, \ldots, U_m)$ from the set $\mathcal{T}$ as follows. Each $U_k$ independently selects a countable sequence of transformations $t_{1,U_k}^{(k)}, t_{2,U_k}^{(k)}, \ldots \in \mathcal{T}^{(k)}$. Then, $t(U_1, \ldots, U_m)$ is defined by interleaving these transformations

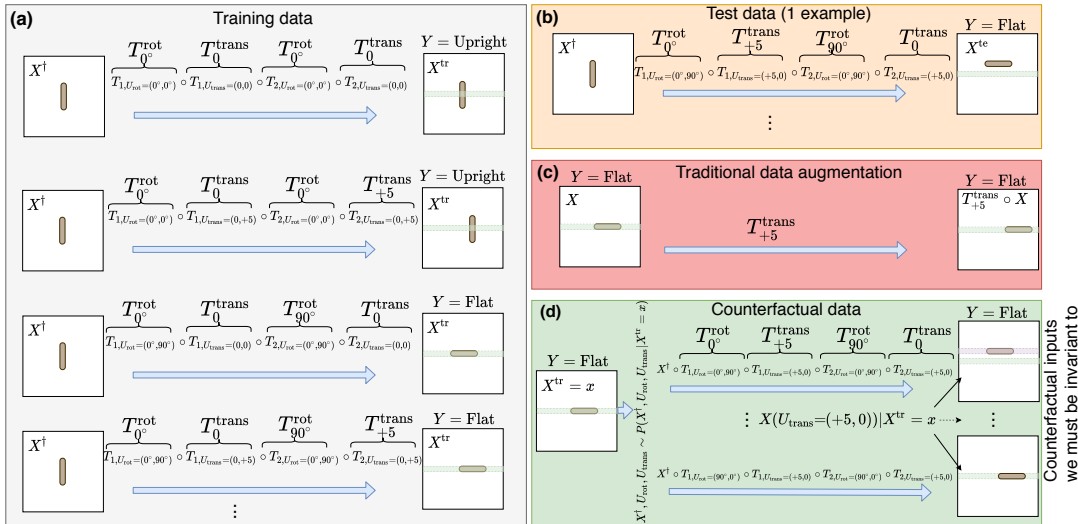

Figure 1: Example that illustrates a few important concepts. **(a)** Training data shows how Equations (2) to (4) define the training distribution $P(X^{\text{tr}}, Y^{\text{tr}})$. Task: Given an image of a rod (shown in brown), we wish to predict the orientation of the rod, i.e., whether the rod is upright or flat ($Y := h(U_{\text{rot}})$). In this example, we have $\mathbb{D} = \{\text{rot}\}$ (image rotations $0°$ and $90°$) and $\bar{\mathbb{D}} = \{\text{trans}\}$ (horizontal translations of $-5, 0, +5$ units) as any horizontal translation does not affect the orientation of the rod. **(b)** The test data (only a single example shown) suffers an OOD shift through a different distribution over $P(U_{\text{trans}})$, where non-zero translations can happen before the second rotation. **(c)** Here we illustrate why an invariance that is good for *traditional data augmentation*, such as counting the brown pixels in the green shaded area, would fail in test if, say, a $+5$ units horizontal translation happens before a rotation. **(d)** Here we illustrate why counterfactual language is needed to define how the input data would change in the presence of changes to $U_{\text{trans}}$. Using counterfactuals, it is finally clear that the invariant representation must be able to also consider the number of brown pixels inside the horizontal purple and green bands (among other horizontal bands).

$t(U_1, \ldots, U_m) := (t^{(1)}_{1,U_1} \circ \cdots \circ t^{(m)}_{1,U_m}) \circ \cdots \circ (t^{(1)}_{r,U_1} \circ \cdots \circ t^{(m)}_{r,U_m}) \circ \cdots$ to construct the path described above. Since $\mathcal{T}^{(1)}, \mathcal{T}^{(2)}, \ldots$ contain the identity transformation, $t(U_1, \ldots, U_m)$ can be described by a finite sequences of transformations. The observed $X$ is the result of a transformation of $X^\dagger$

$$X := t(U_1, \ldots, U_m) \circ X^\dagger \,. \tag{3}$$

Finally, the label $Y$ is defined as a function of the untransformed canonical input $X^\dagger$ as

$$Y := h(X^\dagger, (U_i)_{i \in \mathbb{D}}, U_Y) \,, \tag{4}$$

where $\mathbb{D} \subseteq \{1, \ldots, m\}$ is unknown. This means that $Y$ is not invariant with respect to equivalence relations $\sim_i, i \in \mathbb{D}$, i.e., examples $\boldsymbol{x}$ and $\boldsymbol{x}' \in [\boldsymbol{x}]^{(i)}$ can have different labels. A distribution over the variables $U_u, \{U_i\}_{i=1}^m, U_Y$ entails a joint distribution $P(X, Y)$ over the observed variables.

**Illustrative SCM example.** Figure 1 illustrates our data generation process. The training data Figure 1(a) has $X^\dagger$ defined as a centered upright brown rod (i.e., $X^\dagger$ is deterministic). The label $Y$ is defined by the rotation transformations $\mathcal{T}^{\text{rot}} = \{T^{\text{rot}}_{0°}, T^{\text{rot}}_{90°}\}$. The image can also be horizontally translated by $\{-5, 0, 5\}$ units via transformations $\mathcal{T}^{\text{trans}} = \{T^{\text{trans}}_{-5}, T^{\text{trans}}_0, T^{\text{trans}}_{+5}\}$ (only 0 and $+5$ translations are depicted), but $Y$ does not depend on these horizontal translations. The transformations applied to $X^\dagger$ are randomly chosen via $U_{\text{rot}}$ and $U_{\text{trans}}$, which are two bidimensional vectors indexing a sequence four transformations that interleave rotations and translations (see Figure 1). A representation that counts the number of brown pixels in the green shaded area of $X^{\text{tr}}$ is enough to achieve 100% accuracy in the training distribution. We formally define OOD distribution shifts next using Figure 1 for illustration.

**OOD distribution shift.** Let $\bar{\mathbb{D}} = \{1, \ldots, m\} \backslash \mathbb{D}$ be the complement of the set of symmetry relations $\mathbb{D}$ that $Y$ depends on. We define the OOD distribution shift between train and test as a shift in the distribution of $P((U_i)_{i \in \bar{\mathbb{D}}})$, influencing the distribution of input transformations in Equation (3), which in turn can shift the distributions $P(X^{\text{tr}}), P(Y^{\text{tr}}|X^{\text{tr}}), P(Y^{\text{tr}}, X^{\text{tr}})$ to

$P(X^{\text{te}}), P(Y^{\text{te}}|X^{\text{te}}), P(Y^{\text{te}}, X^{\text{te}})$ respectively. Since $X$ does not causally affect $Y$ in our structural causal model (Equation (4)), changes in input transformations are able to shift $P(Y|X)$. For example, in Figure 1(b) the test data (only a single example shown) could suffer an OOD shift due to a different distribution over $P(U_{\text{trans}})$ that introduces non-zero translations before the second rotation. Note that the representation that counted the number of brown pixels in the green shaded area, which was perfect for the training inputs $X^{\text{tr}}$, will achieve poor accuracy in the test inputs $X^{\text{te}}$.

**Learning OOD classifiers.** Equation (4) shows that the label $Y$ is invariant to changes in the distribution of $(U_i)_{i \in \bar{\mathbb{D}}}$ in the test distribution, but we do not know $\bar{\mathbb{D}}$. Hence, if our representation of $X$ is invariant to changes in the distribution of $(U_i)_{i \in \bar{\mathbb{D}}}$, we will be able to perform the OOD task.

# 4 ASYMMETRY LEARNING & FINDING THE RIGHT REPRESENTATION SYMMETRY FOR THE OOD TASK

## 4.1 FINDING OOD-INVARIANT REPRESENTATIONS AS CAUSAL STRUCTURE DISCOVERY

We first define the process of finding an OOD invariant representations for the symmetries $\{\sim_i\}_{i \in \bar{\mathbb{D}}}$ our classifier should be invariant to in the test data. Since $Y$ does not depend on $\{U_i\}_{i \in \bar{\mathbb{D}}}$, we will make a representation of $X$ that is invariant to transformations driven by $\{U_i\}_{i \in \bar{\mathbb{D}}}$.

Definition 1 introduces the concept of *counterfactual invariance* for symmetry transformations. We note that this definition is less restrictive than the parallel work of Veitch et al. (2021, Definition 1.1): whereas Veitch et al. (2021, Definition 1.1) require invariance over the entire sample space, we only require invariance over the test support of transformation variable $U_i$. The definitions are equivalent if the test support is the entire sample space of $U_i$.

**Definition 1** (Counterfactual-invariant representations for symmetric transformations). *Assume the SCM defined in Equations* (2) *to* (4). *A representation* $\Gamma_i : \mathcal{X} \to \mathbb{R}^d$, $d \geqslant 1$, *is counterfactual-invariant to the transformations* $T_{1,U_i}, T_{2,U_i}, \ldots$ *of equivalence relation* $\sim_i$, $1 \leqslant i \leqslant m$, *if*

$$\Gamma_i(x) = \Gamma_i(X(U_i = \tilde{u}_i)|X = x)$$

*almost everywhere,* $\forall \tilde{u}_i \in \text{supp}(U_i^{te}), \forall x \in \text{supp}(X^{tr})$, *where* $\text{supp}(A)$ *is the support of random variable* $A$. *A representation* $\Gamma_{\mathbb{S}} : \mathcal{X} \to \mathbb{R}^d$, $d \geqslant 1$, *is counterfactual-invariant to a subset* $\mathbb{S} \subseteq \{1, \ldots, m\}$ *if it is jointly counterfactual-invariant to the transformation indices* $\{U_j\}_{j \in \mathbb{S}}$ *of equivalence relations* $\{\sim_j\}_{j \in \mathbb{S}}$.

We refer the reader to Equation (1) for the relationship between the counterfactual variables $X(U_i = \tilde{u})|U_i = u$ and $X(U_i = \tilde{u})|X = x$. Figure 1(d) illustrates why counterfactual language is important for our task: It states that given an input $X^{\text{tr}} = x$ we need to know how it would have been different if we had chosen a different distribution $P(U_{\text{trans}})$ resulting in a different sequence of transformations $T_{1,U_{\text{trans}}}, T_{2,U_{\text{trans}}}$. From Figure 1(c) it is clear that we cannot simply data-augment our training data with translations, since we would think that counting brown pixels in the green shaded area is an invariant representation for $U_{\text{trans}}$.

Up until now we have not imposed restrictions on the types of transformations $\mathcal{T}^{(i)}$, $i = 1, \ldots, m$, we consider in this work. Our next results require imposing conditions on these transformations.

**Definition 2** (Equivalence class lumpability). *The quotient space* $\mathcal{X}/\sim_i$ *is the set of equivalence classes of* $\mathcal{X}$ *with respect to equivalence relation* $\sim_i$, $i = 1, \ldots, m$. *Let* $[\boldsymbol{x}]^{(i)} \in \mathcal{X}/\sim_i$ *be the equivalence class of* $\boldsymbol{x} \in \mathcal{X}$ *with respect to equivalence relation* $\sim_i$. *Then,* $\mathcal{X}/\sim_i$ *is said to be lumpable with respect to a transformation set* $\mathcal{T}$ *if* $\forall [\boldsymbol{x}]^{(i)} \in \mathcal{X}/\sim_i$ *and* $\forall t \in \mathcal{T}$,

$$\exists [\boldsymbol{x}']^{(i)} \in (\mathcal{X}/\sim_i) \ \ s.t. \ \ \boldsymbol{x}^* \in [\boldsymbol{x}]^{(i)} \implies t \circ \boldsymbol{x}^* \in [\boldsymbol{x}']^{(i)}.$$

In words, if the lumpability condition in Definition 2 holds for an equivalence relation $\sim_i$ with respect to a set of transformations $\mathcal{T}$, then every transformation in $\mathcal{T}$ maps all points within an equivalence class $[\boldsymbol{x}]^{(i)} \in \mathcal{X}/\sim$ to points in a another equivalence class $[\boldsymbol{x}']^{(i)} \in (\mathcal{X}/\sim)$. To illustrate the lumpability condition, consider two transformation groups $G_1$ and $G_2$ whose transformations commute, i.e., $\forall (t_1, t_2) \in G_1 \times G_2, t_1 \circ t_2 = t_2 \circ t_1$. Then the equivalence classes imposed by $G_i$, i.e., the orbits $[\boldsymbol{x}]^{(i)} = \{t_i \circ \boldsymbol{x} : \forall t_i \in G_i\}$, are lumpable with respect to the transformations $G_j$, for $i, j \in \{1, 2\}$ and $j \neq i$.

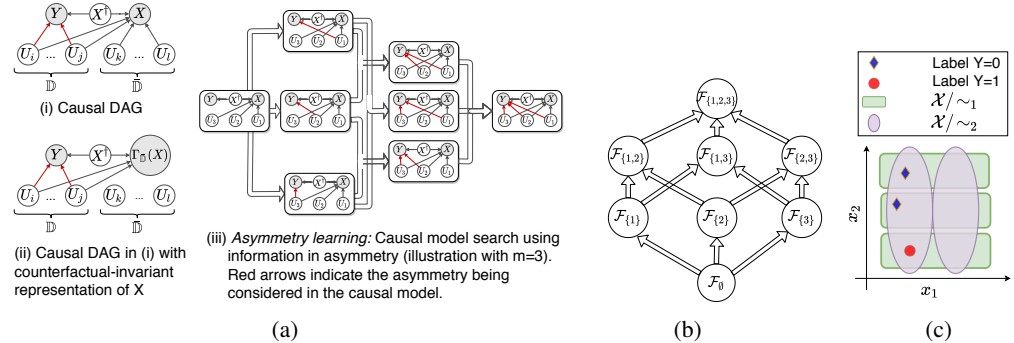

Figure 2: (a) (i) True causal DAG; (ii) causal DAG of counterfactual invariant representation; (iii) Causal model search. (b) Partial order over invariant representations (arrows indicate higher invariance). (c) An example figure where training data has a single example per equivalence class in $\mathcal{X}/\sim_1$ (green rectangles). Then, we have $\text{COMP}(\mathcal{F}_{\{1\}}, \mathcal{D}) = \text{COMP}(\mathcal{F}_{\varnothing}, \mathcal{D})$ even though $\mathcal{F}_{\{1\}}$ is more invariant (simpler) than $\mathcal{F}_{\varnothing}$.

Figure 2a(i) shows our structural causal graph where an edge $U_i \to Y$ exists only if $i \in \mathbb{D}$. Then, we use the definition of lumpability to prove that, under certain conditions, a most-expressive representation $\Gamma_i$ invariant with respect to $\sim_i$ allows us to identify if there is no edge $U_i \to Y$ in the causal DAG.

**Theorem 1** (Counterfactual invariance & causal DAG identification). *Let $\mathcal{X}/\sim_i$ be lumpable given every $\mathcal{T}^{(j)}, j \neq i$ as in Definition 2. Then, the structural causal DAG implied by Equations (2) to (4) (depicted in Figure 2a(i)) does not contain the edge $U_i \to Y$ iff*

$$|P(Y|\Gamma_i(X), U_Y) - P(Y|X, U_Y)|_{TV} = 0, \qquad (5)$$

$\forall P(X^\dagger), \forall P(U_1), \ldots, \forall P(U_m)$, *where $\Gamma_i$ is a most-expressive representation that is invariant with respect to $\sim_i$.*

The proof is in the Appendix. With the lumpability assumption of $\mathcal{X}/\sim_i$, $\Gamma_i$ in Theorem 1 is a counterfactual-invariant representation. We now use Figure 2a(ii) to describe the result in Theorem 1. We first note that the representation $\Gamma_{\bar{\mathbb{D}}}$ depicted in the figure is counterfactual invariant to $\bar{\mathbb{D}}$, and hence also counterfactual invariant to $k \in \bar{\mathbb{D}}$. Next we see that since the representation $\Gamma_{\bar{\mathbb{D}}}$ is counterfactual invariant to $U_k$, there is no arrow $U_k \to \Gamma_{\bar{\mathbb{D}}}(X)$ in Figure 2a(ii). If there is no arrow $U_k \to Y$, the missing arrows from $U_k$ to $\Gamma_{\bar{\mathbb{D}}}(X)$ will have no influence in the ability of $\Gamma_{\bar{\mathbb{D}}}(X)$ to predict $Y$, assuming $\Gamma_{\bar{\mathbb{D}}}$ is most-expressive. If there is an arrow $U_k \to Y$, cutting the arrow $U_k \to \Gamma_{\bar{\mathbb{D}}}(X)$ creates a loss in predictive performance from $\Gamma_{\bar{\mathbb{D}}}(X)$ to $Y$ for some distribution of the background and observable variables. If $\Gamma_{\bar{\mathbb{D}}}(X)$ never loses any predictive power over $Y$ for any distribution of the background and observable variables, then there is no arrow $U_k \to Y$.

**Assumption 1** (Asymmetry learning training data). *In asymmetry learning we assume that every $\mathcal{X}/\sim_i, i \in \{1, \ldots, m\}$ is lumpable given $\mathcal{T}^{(j)}, j \neq i$, and that in a large training dataset sampled from $(Y^{tr}, X^{tr})$, an arrow $U_j \to Y$ in the causal DAG of Figure 2a(i), $j \in \{1, \ldots, m\}$, contains observations of $\{U_j\}_{j \in \mathbb{D}}$ that violate Equation (5). Hence, if Equation (5) holds for some $i \in \{1, \ldots, m\}$ in this dataset, we can conclude that there is no arrow $U_i \to Y$ in the true causal DAG. See Appendix A for a justification of this assumption.*

Next we use Assumption 1 and the previous results to search for the right OOD invariance.

## 4.2 CAUSAL STRUCTURE DISCOVERY OF RELEVANT SYMMETRIES

We need a general procedure for obtaining the unknown set $\mathbb{D}$, which is equivalent to finding all transformations indices $\{U_i\}_{i \in \mathbb{D}} \subseteq \{U_1, \ldots, U_m\}$ that act as confounders between $Y$ and $X$ in the causal DAG in Figure 2a(i). Finding whether an edge exists or not in the causal DAG is known as the causal structure discovery problem (e.g., Heinze-Deml et al. (2017)). The principle of our search is learning the causal structure with the fewest possible edges into $Y$ (i.e., where $Y$ is invariant to most $U_i$, $i = 1, \ldots, m$) while also maximizing the likelihood of the observed data. Accordingly, we take the score-based causal discovery approach (Chickering (2002); Huang et al. (2018)) that assigns scores to each allowed DAG based on the training data and the complexity of the DAG to find a *minimal* causal structure that fits the training data. This idea is visualized in Figure 2a(iii)

where causal graphs with more edges between the transformation indices into $Y$ are defined to have higher complexity and are higher up in the partial ordering. Our search space is simpler than typical structure discovery tasks: The DAGs in our search space have the same structure for $X$ and only differ in edges of the form $U_i \to Y, i \in \{1, \ldots, m\}$. Next, we describe a scoring criterion that uses Theorem 1 and counterfactual-invariant representations to assign scores to the corresponding causal structures.

**Proposed DAG scoring criterion.** For each DAG in the search space, we wish to assign a score based on the training data $\mathcal{D} = \{(\boldsymbol{x}^{(i)}, \boldsymbol{y}^{(i)}\}_{i=1}^{n^{\text{tr}}}$ under Assumption 1 for a classification task with $C$ classes. Theorem 1 shows that there is a correspondence between a causal structure without the edge $U_i \to Y$ and a predictive probability gap between the original input and a most-expressive representation $\Gamma_i$ that is counterfactually-invariant to $U_i$. Thus, under Assumption 1, we can represent the causal search from Figure 2a(iii) in terms of a search over counterfactually-invariant representation function classes as shown in Figures 2a(iii) and 2b. Formally, we are given a collection of function classes $\mathscr{F} := \{\mathcal{F}_{\mathbb{S}} : \mathbb{S} \subseteq \{1, \ldots, m\}\}$, where $\mathcal{F}_{\mathbb{S}}$ is a family of functions $\Gamma_{\mathbb{S}}$ that are counterfactually-invariant to all $U_i, i \in \mathbb{S}$ (Definition 1). We wish to score each of the function classes $\mathcal{F}_{\mathbb{S}} \in \mathscr{F}$ to indirectly learn the correct causal structure.

The minimum description length (MDL) principle (Schwarz, 1978) is commonly used for causal structure discovery (Budhathoki & Vreeken, 2016; 2017) and comes with the key insight that learning from data can be viewed as compressing it. Given the collection $\mathscr{F}$ and the training dataset $\mathcal{D}$, MDL finds the function class $\mathcal{F}_{\mathbb{S}} \in \mathscr{F}$ that compresses $\mathcal{D}$ the most. While there are several ways of encoding a dataset given the function class, normalized maximum likelihood (NML) code is known to be optimal (Shtarkov, 1987). NML code is computed as follows

$$L_{\text{nml}}(\mathcal{F}_{\mathbb{S}}, \mathcal{D}) = -L(\mathcal{F}_{\mathbb{S}}|\mathcal{D}) + \text{COMP}(\mathcal{F}_{\mathbb{S}}, \mathcal{D}) \,, \tag{6}$$

where $L(\mathcal{F}_{\mathbb{S}}|\mathcal{D}) = \sup_{\Gamma_{\mathbb{S}} \in \mathcal{F}_{\mathbb{S}}} \sum_{i=1}^{n^{\text{tr}}} \log P(\boldsymbol{y}^{(i)}|\Gamma_{\mathbb{S}}(\boldsymbol{x}^{(i)}))$ is the maximum log-likelihood of $\mathcal{F}_{\mathbb{S}}$ given the data and

$$\text{COMP}(\mathcal{F}_{\mathbb{S}}, \mathcal{D}) = \log \left[ \sum_{\substack{\boldsymbol{y}^{(1)}, \ldots, \boldsymbol{y}^{(n^{\text{tr}})}: \\ \boldsymbol{y}^{(i)} \in \{0, \ldots, C\}}} \sup_{\Gamma_{\mathbb{S}} \in \mathcal{F}_{\mathbb{S}}} \prod_{i=1}^{n^{\text{tr}}} P(\boldsymbol{y}^{(i)}|\Gamma_{\mathbb{S}}(\boldsymbol{x}^{(i)})) \right] \,, \tag{7}$$

measures the complexity of the function class $\mathcal{F}_{\mathbb{S}}$ by computing how well it can represent different label distributions for the given inputs $\{\boldsymbol{x}^{(i)}\}_{i=1}^{n^{\text{tr}}}$ in training. We can estimate the combinatorial sum in Equation (7) by uniformly sampling random labels for all the training examples.

Since $\text{COMP}(\mathcal{F}_{\mathbb{S}}, \mathcal{D})$ is computed using the training data, it may underestimate the complexity of function classes if, for instance, all the training examples are generated with $U_i = u_i$. Then, $\mathcal{F}_{\{i\}}$ and $\mathcal{F}_{\varnothing}$ are given the same score even though $\mathcal{F}_{\{i\}}$ is clearly more invariant and thus, a simpler function class. This can happen in practice if, say, all images are upright in training with no rotations applied; both rotation-invariant and rotation-sensitive function classes get the same complexity score.

In order to break the above ties of our COMP score, asymmetry learning adds an additional term to the NML score that chooses models that have higher invariance based on the partial order (see Figure 2b). We extend the penalty proposed by Mouli & Ribeiro (2021) and use $R(\mathcal{F}_{\mathbb{S}}) := |\{\mathcal{F}' : \mathcal{F}' \in \mathscr{F}, \mathcal{F}' > \mathcal{F}_{\mathbb{S}}\}|$, the number of function classes that are higher in the partial order than $\mathcal{F}_{\mathbb{S}}$, as the tie-breaking term. For example, in figure $R(\mathcal{F}_{\{1\}}) = |\{\mathcal{F}_{\{1,2\}}, \mathcal{F}_{\{1,3\}}, \mathcal{F}_{\{1,2,3\}}\}| = 3$. We define the final score of each function class $\mathcal{F}_{\mathbb{S}} \in \mathscr{F}$ as

$$S(\mathcal{F}_{\mathbb{S}}, \mathcal{D}) = L_{\text{nml}}(\mathcal{F}_{\mathbb{S}}, \mathcal{D}) + R(\mathcal{F}_{\mathbb{S}}) \,. \tag{8}$$

The score in Equation (8) can be minimized by a score-based causal discovery algorithm to obtain the final DAG. We use Greedy Equivalence Search (Chickering, 2002) to showcase a concrete instantiation of asymmetry learning. Other score-based structure discovery algorithms could also be used.

**Greedy Equivalence Search.** Greedy Equivalence Search (GES) is a greedy search algorithm that optimizes a given scoring function over DAGs. In our setting, the search begins with a DAG with no edges of the form $U_i \to Y, i \in \{1, \ldots, m\}$. In the first phase, GES adds these edges one at a time

Table 1: Results for different function classes on the pendulum task with $\mathbb{D} = \{1\}$ and $\mathbb{D} = \{1,2\}$. $R(\mathcal{F})$, $\widehat{\text{COMP}}(\mathcal{F}, \mathcal{D})$ and $S(\mathcal{F}, \mathcal{D})$ are discussed as in Section 4.2. Bold values indicate the function class chosen by GES method with the proposed scoring criterion. Test accuracy is computed on the extrapolated dataset after shifting the distribution of $P(\{U_i\}_{i \in \bar{\mathbb{D}}})$.

| Model class | Architecture | $R(\mathcal{F})$ | $\mathbb{D} = \{1\}$ | | | | $\mathbb{D} = \{1,2\}$ | | | |
| | | | $\widehat{\text{COMP}}(\mathcal{F},\mathcal{D})$ | $S(\mathcal{F},\mathcal{D})$ | Train Acc. | Test Acc. | $\widehat{\text{COMP}}(\mathcal{F},\mathcal{D})$ | $S(\mathcal{F},\mathcal{D})$ | Train Acc. | Test Acc. |
|---|---|---|---|---|---|---|---|---|---|---|
| $\mathcal{F}_2$ | $X \to z_1 \to Y$ | 0 | **0.282** | **23.89** | **98.5 (0.9)** | **98.3 ( 1.4)** | 0.501 | 532.84 | 72.7 (0.4) | 69.4 (0.5) |
| $\mathcal{F}_1$ | $X \to z_2 \to Y$ | 0 | 0.382 | 633.32 | 63.8 (7.0) | 51.2 ( 1.0) | 0.292 | 284.75 | 85.2 (0.5) | 84.6 (0.2) |
| $\mathcal{F}_\varnothing$ | $X \to Y$ | 2 | 1.256 | 26.80 | 98.9 (0.8) | 77.6 (11.5) | **0.995** | **4.54** | **99.7 (0.2)** | **99.5 (0.2)** |

that maximally improve the score in Equation (8) until there is no improvement. In the second phase, GES begins from the DAG obtained at the end of first phase and deletes edges one at a time until such deletions do not improve the score. The DAG obtained at the end of the second phase is the final output of the algorithm. Under the causal Markov and faithfulness assumptions, Chickering (2002) showed that GES is optimal in the large sample limit if the scoring function is locally consistent.

## 5 RESULTS

**Pendulum task description.** We evaluate the proposed method in a simulated classification task. Our input $x$ is a motion vector over time $(\theta_t, \frac{d\theta_t}{dt})_{t=1}^T$ of a simple pendulum of an unknown length $l$ after it is dropped from some initial angle $\theta_0$ with $\frac{d\theta_0}{dt} = 0$. After an initial $\tau$ seconds of uninterrupted motion, we simulate an elastic collision by placing another object of same mass at the bottom. The classification task is to predict whether the kinetic energy imparted by the pendulum is enough to move the second object beyond a certain threshold.

*Physical properties and equivalence relations.* We consider the following two properties of the dynamical system described above: $z_1 : \mathcal{X} \to \mathbb{R}$ which computes the initial potential energy of the system and $z_2 : \mathcal{X} \to \mathbb{R}$ which returns the time of collision. The equivalence relations $\sim_1$ and $\sim_2$ are defined using these properties as defined in Section 2. For instance, two pendulum motion curves $x, x'$ are equivalent with respect to $\sim_1$, i.e., $x \sim_1 x'$, if they have the same time of collision, $z_2(x) = z_2(x')$. Then $\mathcal{T}^{(1)}$ consists of transformations that change the initial potential energy of the system (for example, by changing the length of the pendulum or the initial dropping angle $\theta_0$) while keeping the time of collision same. Similarly, $x \sim_2 x'$ if their respective potential energies are the same and transformations in $\mathcal{T}^{(2)}$ change the time of collision while keeping the same initial potential energies. Note that the space of equivalence classes $\mathcal{X} / \sim_1$ is lumpable with respect to $\mathcal{T}^{(2)}$ and vice versa (Definition 2). Thus, by Theorem 1, we can use predictive performance of counterfactual-invariant representations for scoring the causal DAGs.

*Unknown $\mathbb{D}$ and OOD classification.* We consider two scenarios for the label $Y$ given $X$. First, if the motion of the pendulum is not damped by friction, then $Y$ depends only on $z_1(x)$, i.e, $\mathbb{D} = \{1\}$. Second, if the motion of the pendulum is damped, then $Y$ depends on both $z_1(x)$ and $z_2(x)$, i.e., $\mathbb{D} = \{1, 2\}$. The extrapolation test data is generated by shifting the distribution of the background variables $\{U_i\}_{i \in \bar{\mathbb{D}}}$. The task of a structure discovery algorithm is to correctly identify $\mathbb{D}$.

*Results.* We use the greedy equivalence search (GES, Section 4.2) algorithm to search over the different causal graphs with the proposed scoring criterion defined in Equation (8). We build classes of counterfactual-invariant representations $\mathcal{F}_{\mathbb{S}}$ corresponding to each possible value of $\mathbb{S} \subsetneq \{1, 2\}$, where every $\Gamma_{\mathbb{S}} \in \mathcal{F}_{\mathbb{S}}$ is invariant to $\{U_i\}_{i \in \mathbb{S}}$. For example, $\mathcal{F}_{\{1\}}$ is a family of feedforward neural networks that only take $z_2(x)$ as input, i.e., invariant to $z_1(x)$, whereas $\mathcal{F}_\varnothing$ is a sequence model (e.g., LSTM) with no invariance. Table 1 reports the estimated complexity $\widehat{\text{COMP}}(\mathcal{F}, \mathcal{D})$ and the final scores $S(\mathcal{F}, \mathcal{D})$ for the different function classes for the two tasks. The bold values indicate the function class chosen by the GES algorithm. When $\mathbb{D} = \{1\}$, the greedy search stops after adding the edge $U_1 \to Y$ as adding the second edge $U_2 \to Y$ only worsens (increases) the score. When $\mathbb{D} = \{1, 2\}$, the greedy search is able to improve the score by adding both edges, first $U_1 \to Y$ and then $U_2 \to Y$. In both the cases, the extrapolation test accuracy achieved by the chosen model class is the highest.

**Image classification task.** Appendices A.4 and A.5 also offers an application to image classification using image transformation sets (groups and nongroups).

## 6 RELATED WORK

**Counterfactual inference and invariances.** Recent efforts have brought causal inference to machine learning (extensively reviewed in Schölkopf et al. (2021); Schölkopf (2022)). Invariant Causal Prediction (Peters et al., 2015; Heinze-Deml et al., 2018) and Invariant Risk Minimization methods (Arjovsky et al., 2019; Bellot & van der Schaar, 2020) learn representations that are invariant across multiple environments but have been shown to be insufficient for OOD generalization in classification tasks without additional assumptions Ahuja et al. (2021). Wang & Jordan (2021) use counterfactual language to formally define and learn non-spurious representations from a single environment that can extrapolate to new environments. Veitch et al. (2021) define counterfactual invariant predictors $f(X)$ when $X$ has a single parent $Z$ and provide conditions such predictors must satisfy over the observed distribution (given an SCM). Kaushik et al. (2020; 2021) propose counterfactual data augmentation for text datasets but they either require a fully-specified toy SCM or rely on humans-in-the-loop to generate the counterfactual data. Other counterfactual methods (Johansson et al., 2016; Shalit et al., 2017; Qidong et al., 2020) learn representations to predict counterfactual change in some observed variables whereas in our setting, the transformation variables $U_i$ that generate the observed $X$ are unobserved. In-depth comparison of our work with the existing counterfactual methods is presented in Appendix A.3.

**Domain adaptation and domain generalization.** Domain adaptation and domain generalization (e.g. (Long et al., 2017; Muandet et al., 2013; Quionero-Candela et al., 2009; Rojas-Carulla et al., 2018; Shimodaira, 2000; Zhang et al., 2015) and others) consider observed or known shifts in the data distribution, for instance, given the test distribution $P(X^{\text{te}})$, rather than counterfactual questions.

**Causal structure discovery.** The methods for causal structure discovery can be broadly classified into two categories. Constraint-based approaches (e.g., Spirtes et al. (2001); Sun et al. (2007)) use conditional independence tests and reject causal graphs that impose more independence than what is observed in data. On the other hand, score-based causal discovery approaches (e.g., Chickering (2002); Huang et al. (2018); Ding et al. (2020); Zhu et al. (2020)) assign scores to each allowed causal graph based on the data and find the one with best score. While there are several works (Budhathoki & Vreeken, 2016; 2017; Bornschein et al., 2021) that use minimum description length (MDL) (Schwarz, 1978) as a scoring criterion, we show why it is insufficient for out-of-distribution tasks and use an additional term for tie-breaking. Goudet et al. (2017) minimize the divergence between a distribution generated by a learnt causal DAG and the observed data distribution; however the method is limited to orienting edges over observed variables, whereas our transformation variables $U_i$ are unobserved. Recently, GFlowNets Bengio et al. (2021a;b) have been used to sample DAGs proportional to a score function for Bayesian structure learning Deleu et al. (2022), however we are interested in finding the best DAG with the minimum score.

**Group-invariant representations.** Majority of the works strictly enforce G-invariances either within the architecture (e.g., Zaheer et al. (2017); Cohen et al. (2016); Lyle et al. (2020); Murphy et al. (2019a)) or via data-augmentation (Chen et al., 2020) and do not handle the case when the target is actually influenced by the transformation of the input. Other works (Benton et al., 2020; Zhou et al., 2020; van der Wilk et al., 2018; Anselmi et al., 2019) consider learning symmetries from the training data but do not consider the extrapolation task that we show can be solved only under certain conditions. Mouli & Ribeiro (2021) consider the special case where the transformations are from normal subgroups and do not formally describe the causal task. These works rely on invertible transformations while we define symmetries more generally via equivalence relations. Dubois et al. (2021) also define invariances via equivalence relations and, under the assumption that all such invariances hold in the data, the authors design methods for data compression. Our goal is rather different: We want to discover which equivalence relations (transformations thereof) affect the label.

## 7 CONCLUSIONS

This work considered an out-of-distribution (OOD) classification task where the shift between train and test environments is through different symmetry transformations of the input, where symmetry transformations are defined via equivalence relations over the input space. We described the task of finding symmetries that affect the label as a causal structure discovery task and show that, under certain conditions, we can use the predictive performance of invariant representations on the observational data to predict whether an edge exists in the causal DAG (Theorem 1). We then proposed an MDL-based scoring for this causal structure discovery. Finally, we test our approach in two simulated physics tasks and six image classification tasks.

ACKNOWLEDGMENTS

This work was funded in part by the National Science Foundation (NSF) Awards CAREER IIS-1943364 and CCF-1918483, the Purdue Integrative Data Science Initiative, and the Wabash Heartland Innovation Network. Any opinions, findings and conclusions or recommendations expressed in this material are those of the authors and do not necessarily reflect the views of the sponsors.

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

# A Appendix

## A.1 Justification for Assumption 1.

The above assumption is inspired by the deep relationship between symmetries and intelligence. Young children, unlike monkeys and baboons, assume that a conditional stimulus F given another stimulus D extrapolates to a symmetric relation D given F without ever seeing any such examples (Sidman et al., 1982). That is, if given D, action F produces a treat, the child assumes that given F, action D also produces a treat. Young children differ from primates in their ability to use symmetries to build conceptual relations beyond visual patterns (Sidman & Tailby, 1982; Westphal-Fitch et al., 2012), allowing extrapolations from intelligent reasoning. However, forcing symmetries against data evidence is undesirable, since symmetries can provide valuable information when they are broken. Unsurprising, humans are generally able to quickly find and pay attention to some types of asymmetries.

## A.2 Proof of Theorem 1

**Theorem 1** (Counterfactual invariance & causal DAG identification). *Let $\mathcal{X}/\sim_i$ be lumpable given every $\mathcal{T}^{(j)}, j \neq i$ as in Definition 2. Then, the structural causal DAG implied by Equations (2) to (4) (depicted in Figure 2a(i)) does not contain the edge $U_i \to Y$ iff*

$$|P(Y|\Gamma_i(X), U_Y) - P(Y|X, U_Y)|_{TV} = 0, \tag{5}$$

$\forall P(X^\dagger), \forall P(U_1), \dots, \forall P(U_m)$*, where $\Gamma_i$ is a most-expressive representation that is invariant with respect to $\sim_i$.*

*Proof.* Notation (following Equation (3)): The observed input $X$ is $X := t(U_1, \dots, U_{i-1}, u_i, U_{i+1}, \dots, U_m) \circ X^\dagger$ where $t(U_1, \dots, U_m)$ is obtained by interleaving the transformation sequences from each individual $U_1, \dots, U_m$ and we have set $U_i = u_i$.

Necessity: We wish to show that if the SCM does not contain edge $U_i \to Y$, then Equation (5) holds for all $P(X^\dagger), P(U_1), \dots, P(U_m)$. By this assumption, $Y$ outputs the same label for any value of $U_i$. Consider the collection of equivalence classes $\mathcal{X}/\sim_i$. By the lumpability condition of Definition 2, all transformations $t^{(j)} \in \mathcal{T}^{(j)}, j \neq i$, map all points in one equivalence class of $\sim_i$ to points in a different one. On the other hand, all transformations $t^{(i)} \in \mathcal{T}^{(i)}$ map points to other points within the same equivalence class under $\sim_i$. Now, consider the equivalence class of $X$ after all the transformations have been applied to $X^\dagger$. The equivalence class of $X = t(U_1, \dots, U_{i-1}, u_i, U_{i+1}, \dots, U_m) \circ X^\dagger$ is the same as that of $X^* = t(U_1, \dots, U_{i-1}, u_i^{id}, U_{i+1}, \dots, U_m) \circ X^\dagger$ where $U_i = u_i^{id}$ always selects identity transformations. This is because changing $u_i$ to $u_i^{id}$ only impacts the transformations chosen from $\mathcal{T}^{(i)}$, and these transformations do not change the equivalence class under $\sim_i$. Thus, we have shown that we reach the same equivalence class under $\sim_i$ for both $X$ and $X^*$.

Now let $\Gamma_i$ be a most-expressive representation that is invariant with respect to $\sim_i$. By definition, $\Gamma_i$ outputs the same value within an equivalence class, thus, $\Gamma_i(X) = \Gamma_i(X^*)$. But since by assumption $U_i \to Y$ does not exist, $X$ and $X^*$ have the same label always. Thus, there is no loss of information incurred by $\Gamma_i$ in predicting $Y$ with the additional restraint $\Gamma_i(X) = \Gamma_i(X^*)$. Since $\Gamma_i$ is most-expressive, we have $P(Y = y|\Gamma_i(X), U_Y) = P(Y = y|X, U_Y)$ for all $y \in \mathcal{Y}$. This holds for all values of $u_i$, and hence we get the desired result for any distribution $P(U_i)$.

Sufficiency: We wish to show that if Equation (5) holds for all $P(X^\dagger)$ and $P(U_1), \dots, P(U_m)$, then there is no edge $U_i \to Y$ in the causal graph. We will prove by contrapositive: Assume there is an edge $U_i \to Y$, then we will show there exists distributions $P(X^\dagger)$ and $P(U_1), \dots, P(U_m)$ such that Equation (5) does not hold.

Define $P(X^\dagger) = \delta_{x^\dagger}$ for some $x^\dagger \in \mathcal{X}$ where $\delta$ denotes a Dirac-delta function. Define $P(U_i = u_i^{id}) = 0.5$ and $P(U_i = u_i) = 0.5$ for $u_i^{id}, u_i \in \text{supp}(U_i)$. As usual, $u_i^{id}$ always selects the identity transformation, and $u_i$ selects a single transformation $t_{u_i} \in \mathcal{T}^{(i)}$. Similarly, for all $j \neq i$, define $P(U_j) = \delta_{u_j^{id}}$ for $u_j^{id} \in \text{supp}(U_j)$ that only select identity transformations. Now, there are two possible observed inputs: $\boldsymbol{x} = t(u_1^{id}, \dots, u_m^{id}) \circ x^\dagger = x^\dagger$ and $\boldsymbol{x}' = t(u_1^{id}, \dots, u_i, \dots, u_m^{id}) \circ x^\dagger = t_{u_i} \circ x^\dagger$. Finally, define $Y := \mathbf{1}(U_i = u_i^{id})$, thus $\boldsymbol{x}$ and $\boldsymbol{x}'$ have different labels. But, any invariant

representation $\Gamma_i$ by definition has $\Gamma_i(\boldsymbol{x}) = \Gamma_i(\boldsymbol{x}')$ since they belong to the same equivalence class. Thus, even if $\Gamma_i$ is most-expressive, we have $|P(Y|\Gamma_i(X), U_Y) - P(Y|X, U_Y)|_{\text{TV}} = 0.5$.

$\square$

### A.3 ADDITIONAL RELATED WORK

**Counterfactual invariances.** Wang & Jordan (2021) use counterfactual language to formally define and learn non-spurious, disentangled representations from a single environment. Our work is different in the following ways. In the structural causal model (SCM) of their work, the authors assume that there are no confounders between the observed $X$ and the label $Y$. However, in our SCM (Figure 2a(i)), we allow unobserved confounders $X^\dagger$ and $U_i, i \in \mathbb{D}$. The hidden transformation variables $U_i, i \in \mathbb{D}$ are confounders because they affect both the observed input $X$ and the labels $Y$. We leverage the fact that the confounders are related to symmetries (and do not affect $X$ arbitrarily) to resolve the issue with unobserved confounding. Wang & Jordan (2021) also require pinpointability of the cause of the observed $X$. In our setting, this is typically not possible since there are multiple paths of transformations from $X^\dagger$ to the same observed $X$. Thus, all the parents of $X$ may not be pinpointable, specifically the transformation variables $U_1, \ldots, U_m$.

Kaushik et al. (2020; 2021) propose counterfactual data augmentation for text datasets where human annotators are asked to make minimal modifications to the input document so as to change its label (for example, by changing a few positive words to negative words) while keeping style, etc. fixed. This type of augmentation essentially asks the labelers to identify all the causal features in the document and make modifications to those features alone. This can be seen as obtaining new counterfactual examples by simulating the causal model and requires knowing the true function that describes how the features affect the labels. We consider the more realistic setting where we do not have access to such a collection of counterfactual examples. In this work, we consider the traditional automated data augmentations under a mostly unknown data generation process, as opposed to the counterfactual data augmentation (Kaushik et al., 2020) that either considers a fully-specified toy SCM or relies on humans-in-the-loop to generate counterfactual data.

In Figure 1(c) we show that the standard data augmentation is not sufficient for the OOD task. However, if one had access to the fully-specified causal model, one could generate the counterfactual data shown in Figure 1(d) and learn an OOD classifier with the counterfactually augmented data (as done by Kaushik et al. (2020)). But our work does not assume access to these counterfactual examples. Additionally, we prove that a counterfactual invariant classifier can be constructed from traditional data augmentation alone if the lumpability condition (Definition 2) is satisfied. This is not the case in Figure 1(d).

Veitch et al. (2021) define counterfactual invariant predictors $f(X)$ when $X$ has a single parent $Z$ and provide conditions such predictors must satisfy over the observed distribution (given an SCM). Note also that Veitch et al. (2021) assume that part of the observed input $X$ ($X_Z^\perp$) is not causally influenced by the confounder $Z$. In our scenarios this is not generally true. For example, under a color change, the entire observed image $X$ changes. Still, we show that the notion of a counterfactual invariant predictor exists. Hence, the definition of Veitch et al. (2021, Lemma 3.1) of a counterfactually invariant predictor that requires a segment of $X$ to not causally depend on $Z$, a fundamental result of their work, unfortunately does not apply to our setting (since $X$ may have no such segment).

### A.4 MNIST-$\{3, 4\}$ EXPERIMENTS WITH FINITE TRANSFORMATION GROUPS

We test our proposed method on out-of-distribution tasks on images where the equivalence relations (symmetries) are provided as transformation groups (e.g., $90°$ rotations). We use the MNIST-$\{3, 4\}$ (colored) dataset (Mouli & Ribeiro, 2021) that only contains digits 3 and 4, and follow their experimental setup. MNIST-$\{3, 4\}$ is used to avoid any confounding factors while testing if the proposed method can learn the correct invariances, not for any practical considerations (e.g., rotated 6 is a 9 and would interfere with some experiments, etc.).

We consider equivalence relations obtained from 3 different transformation groups: rotations by $90°$ (denoted $G_{\text{rot}}$), vertically flipping the image (denoted $G_{\text{v-flip}}$), and permuting the RGB color channels of the image (denoted $G_{\text{col}}$). The 3 corresponding equivalence relations are lumpable (Definition 2) with respect to the transformations in the other two groups in almost all the cases. Only exception

Table 2: Results for different function classes on the MNIST-$\{3, 4\}$ classification task with $\bar{\bar{\mathbb{D}}}$ = $\{$rot, col, vflip$\}$, $\mathbb{D} = \varnothing$, i.e., task is invariant to 3 groups ($\bar{\bar{\mathbb{D}}}$) and sensitive to none ($\mathbb{D}$). $R(\mathcal{F})$, $\widehat{\text{COMP}}(\mathcal{F}, \mathcal{D})$ and $S(\mathcal{F}, \mathcal{D})$ are as discussed in Section 4.2. **Bold** values indicate the function class chosen by GES method with the proposed scoring criterion. Test accuracy is computed on the extrapolated dataset after shifting the distribution of $P(\{U_i\}_{i\in\bar{\bar{\mathbb{D}}}})$. We see that the $S(\mathcal{F}, \mathcal{D})$ loss selects the correct model class in training.

| Model class | $R(\mathcal{F})$ | $+ \widehat{\text{COMP}}(\mathcal{F}, \mathcal{D})$ | $+ \text{NLL}(\mathcal{F}, \mathcal{D})$ | $= S(\mathcal{F}, \mathcal{D})$ | Train Acc | Test Acc |
|---|---|---|---|---|---|---|
| $\mathcal{F}_{\{\}}$ | 7 | 6639.310 | 0.013 | 6646.324 | 100.00 ( 0.00) | 48.38 ( 5.22) |
| $\mathcal{F}_{\{\text{vflip}\}}$ | 3 | 6639.241 | 0.079 | 6642.320 | 100.00 ( 0.00) | 47.08 ( 5.34) |
| $\mathcal{F}_{\{\text{col}\}}$ | 3 | 6639.241 | 0.029 | 6642.270 | 100.00 ( 0.00) | 53.92 ( 2.47) |
| $\mathcal{F}_{\{\text{col,vflip}\}}$ | 1 | 6639.241 | 0.099 | 6640.340 | 100.00 ( 0.00) | 53.15 ( 1.83) |
| $\mathcal{F}_{\{\text{rot}\}}$ | 3 | 6639.241 | 0.037 | 6642.278 | 100.00 ( 0.00) | 53.06 (10.00) |
| $\mathcal{F}_{\{\text{rot,vflip}\}}$ | 1 | 6639.241 | 0.580 | 6640.821 | 100.00 ( 0.01) | 54.86 (13.60) |
| $\mathcal{F}_{\{\text{rot,col}\}}$ | 1 | 6639.241 | 0.043 | 6640.284 | 100.00 ( 0.00) | 90.29 ( 6.76) |
| $\mathcal{F}_{\{\text{rot,col,vflip}\}}$ | **0** | **6639.241** | **0.210** | **6639.451** | **100.00 ( 0.00)** | **92.02 ( 2.99)** |

Table 3: Results for different function classes on the MNIST-$\{3, 4\}$ classification task with $\bar{\bar{\mathbb{D}}} = \{$rot, vflip$\}$, $\mathbb{D} = \{$col$\}$, i.e., task is invariant to rotation and vertical flip groups ($\bar{\bar{\mathbb{D}}}$) but sensitive to color ($\mathbb{D}$). $R(\mathcal{F})$, $\widehat{\text{COMP}}(\mathcal{F}, \mathcal{D})$ and $S(\mathcal{F}, \mathcal{D})$ are as discussed in Section 4.2. **Bold** values indicate the function class chosen by GES method with the proposed scoring criterion. Test accuracy is computed on the extrapolated dataset after shifting the distribution of $P(\{U_i\}_{i\in\bar{\bar{\mathbb{D}}}})$. We see that the $S(\mathcal{F}, \mathcal{D})$ loss selects the correct model class in training.

| Model class | $R(\mathcal{F})$ | $+ \widehat{\text{COMP}}(\mathcal{F}, \mathcal{D})$ | $+ \text{NLL}(\mathcal{F}, \mathcal{D})$ | $= S(\mathcal{F}, \mathcal{D})$ | Train Acc | Test Acc |
|---|---|---|---|---|---|---|
| $\mathcal{F}_{\{\}}$ | 7 | 6639.241 | 0.010 | 6646.251 | 100.00 ( 0.00) | 54.79 ( 0.74) |
| $\mathcal{F}_{\{\text{vflip}\}}$ | 3 | 6639.241 | 0.012 | 6642.253 | 100.00 ( 0.00) | 55.05 ( 1.56) |
| $\mathcal{F}_{\{\text{col}\}}$ | 3 | 6639.240 | 8269.480 | 14911.720 | 41.98 ( 5.79) | 18.81 ( 2.94) |
| $\mathcal{F}_{\{\text{col,vflip}\}}$ | 1 | 6639.241 | 8275.716 | 14915.957 | 42.71 ( 4.07) | 18.62 ( 2.25) |
| $\mathcal{F}_{\{\text{rot}\}}$ | 3 | 6638.946 | 0.132 | 6642.078 | 100.00 ( 0.00) | 91.40 ( 3.19) |
| $\mathcal{F}_{\{\text{rot,vflip}\}}$ | **1** | **6638.428** | **0.504** | **6639.932** | **100.00 ( 0.00)** | **92.32 ( 1.84)** |
| $\mathcal{F}_{\{\text{rot,col}\}}$ | 1 | 6639.241 | 8412.954 | 15053.194 | 37.20 ( 1.97) | 29.25 ( 5.18) |
| $\mathcal{F}_{\{\text{rot,col,vflip}\}}$ | 0 | 6639.239 | 8389.719 | 15028.958 | 38.01 ( 2.02) | 29.98 ( 3.96) |

is the equivalence relation $\sim_{\text{v-flip}}$, which is not lumpable with respect to the transformations in $G_{\text{rot}}$. Consequently, we do not consider a task with invariance to vertical flip alone. We test our method on the same 4 classification tasks proposed by Mouli & Ribeiro (2021) where each task represents the case where the target $Y$ has different invariances, i.e., invariant to all three groups, to two, to one, invariant to none (the task is sensitive to the remaining groups).

We use the VGG architecture (Simonyan & Zisserman, 2014) for image classification and construct a collection of function classes $\mathscr{F} := \{\mathcal{F}_{\mathbb{S}} : \mathbb{S} \subseteq \{\text{rot, col, v-flip}\}\}$ corresponding to various invariant representations. For example, $\mathcal{F}_{\{\text{rot,col}\}}$ is a space of functions (CNNs) that are G-invariant to the rotation and color-permutation groups ($G_{\text{rot}}$ and $G_{\text{col}}$), and $\mathcal{F}_{\varnothing}$ is the space of functions with no invariance (standard CNN).

**Results.** Our results are shown in Tables 2 to 5 for the four tasks respectively where the label is **(i)** invariant to all three groups, **(ii)** invariant to only rotation and vertical flips, **(iii)** invariant to color-permutation, and **(iv)** invariant to none. We show the values for $R(\mathcal{F})$, $\widehat{\text{COMP}}(\mathcal{F}, \mathcal{D})$ and $S(\mathcal{F}, \mathcal{D})$ as as discussed in Section 4.2. Bold values in the tables indicate the function class chosen by GES method with the proposed scoring criterion (minimizing $S(\mathcal{F}, \mathcal{D})$). Test accuracy is computed on the extrapolated dataset after shifting the distribution of $P(\{U_i\}_{i\in\bar{\bar{\mathbb{D}}}})$ (i.e., by applying the transformations that the label is invariant to).

In Tables 2 and 3, we see that the proposed method selects the correct model class in training and achieves the best OOD test accuracy. In Tables 4 and 5, the method is excessively invariant (to vertical flip) but still achieves within 1% of the best OOD test accuracy. The OOD test accuracy of a

Table 4: Results for different function classes on the MNIST-$\{3, 4\}$ classification task with $\bar{\mathbb{D}} = \{\text{col}\}, \mathbb{D} = \{\text{rot}, \text{vflip}\}$, i.e., task is invariant to color ($\bar{\mathbb{D}}$) but sensitive to rotation and vertical flips ($\mathbb{D}$). $R(\mathcal{F}), \widehat{\text{COMP}}(\mathcal{F}, \mathcal{D})$ and $S(\mathcal{F}, \mathcal{D})$ are as discussed in Section 4.2. **Bold** values indicate the function class chosen by GES method with the proposed scoring criterion. Test accuracy is computed on the extrapolated dataset after shifting the distribution of $P(\{U_i\}_{i \in \bar{\mathbb{D}}})$. We see that the $S(\mathcal{F}, \mathcal{D})$ loss selects a model that is excessively invariant in training, but the test accuracy is not that much penalized by the extra invariance (vertical flips).

| Model class | $R(\mathcal{F})$ | $+ \widehat{\text{COMP}}(\mathcal{F}, \mathcal{D})$ | $+ \text{NLL}(\mathcal{F}, \mathcal{D})$ | $= S(\mathcal{F}, \mathcal{D})$ | Train Acc | Test Acc |
|---|---|---|---|---|---|---|
| $\mathcal{F}_{\{\}}$ | 7 | 6639.241 | 2.395 | 6648.636 | 100.00 ( 0.01) | 16.87 ( 5.88) |
| $\mathcal{F}_{\{\text{vflip}\}}$ | 3 | 6639.233 | 5.370 | 6647.603 | 99.99 ( 0.05) | 15.71 ( 5.53) |
| $\mathcal{F}_{\{\text{col}\}}$ | 3 | 6639.196 | 2.315 | 6644.512 | 100.00 ( 0.00) | 97.28 ( 0.28) |
| $\mathcal{F}_{\{\text{col,vflip}\}}$ | **1** | **6639.240** | **3.098** | **6643.337** | **100.00 ( 0.00)** | **96.82 ( 0.54)** |
| $\mathcal{F}_{\{\text{rot}\}}$ | 3 | 6639.228 | 5296.755 | 11938.984 | 56.17 ( 3.90) | 6.20 ( 0.86) |
| $\mathcal{F}_{\{\text{rot,vflip}\}}$ | 1 | 6639.221 | 5325.008 | 11965.228 | 55.96 ( 5.39) | 7.24 ( 1.48) |
| $\mathcal{F}_{\{\text{rot,col}\}}$ | 1 | 6639.218 | 5322.015 | 11962.233 | 56.14 ( 3.31) | 47.98 ( 1.34) |
| $\mathcal{F}_{\{\text{rot,col,vflip}\}}$ | 0 | 6639.230 | 5342.805 | 11982.035 | 55.32 ( 3.80) | 49.25 ( 3.09) |

Table 5: Results for different function classes on the MNIST-$\{3, 4\}$ classification task with $\bar{\mathbb{D}} = \varnothing, \mathbb{D} = \{\text{rot}, \text{col}, \text{vflip}\}$, i.e., task is sensitive to all three groups ($\mathbb{D}$) and insensitive to none ($\bar{\mathbb{D}}$). $R(\mathcal{F}), \widehat{\text{COMP}}(\mathcal{F}, \mathcal{D})$ and $S(\mathcal{F}, \mathcal{D})$ are as discussed in Section 4.2. **Bold** values indicate the function class chosen by GES method with the proposed scoring criterion. Test accuracy is computed on the extrapolated dataset after shifting the distribution of $P(\{U_i\}_{i \in \bar{\mathbb{D}}})$. We see that the $S(\mathcal{F}, \mathcal{D})$ loss selects a model that is excessively invariant in training, but the test accuracy is not that much penalized by the extra invariance (vertical flip).

| Model class | $R(\mathcal{F})$ | $+ \widehat{\text{COMP}}(\mathcal{F}, \mathcal{D})$ | $+ \text{NLL}(\mathcal{F}, \mathcal{D})$ | $= S(\mathcal{F}, \mathcal{D})$ | Train Acc | Test Acc |
|---|---|---|---|---|---|---|
| $\mathcal{F}_{\{\}}$ | 7 | 6639.165 | 1.195 | 6647.360 | 100.00 ( 0.00) | 96.00 ( 0.60) |
| $\mathcal{F}_{\{\text{vflip}\}}$ | **3** | **6639.117** | **3.548** | **6645.665** | **100.00 ( 0.00)** | **95.18 ( 0.45)** |
| $\mathcal{F}_{\{\text{col}\}}$ | 3 | 6639.192 | 7536.167 | 14178.359 | 58.77 ( 3.34) | 32.45 ( 2.18) |
| $\mathcal{F}_{\{\text{col,vflip}\}}$ | 1 | 6639.184 | 7902.462 | 14542.645 | 52.50 ( 7.64) | 31.21 ( 2.48) |
| $\mathcal{F}_{\{\text{rot,col}\}}$ | 1 | 6639.088 | 13628.356 | 20268.443 | 23.78 ( 2.25) | 15.93 ( 0.71) |
| $\mathcal{F}_{\{\text{rot}\}}$ | 3 | 6639.153 | 5259.957 | 11902.110 | 58.12 ( 4.05) | 47.23 ( 1.89) |
| $\mathcal{F}_{\{\text{rot,vflip}\}}$ | 1 | 6639.827 | 5267.771 | 11908.598 | 57.13 ( 1.38) | 47.57 ( 2.15) |
| $\mathcal{F}_{\{\text{rot,col,vflip}\}}$ | 0 | 6639.055 | 13705.123 | 20344.178 | 22.97 ( 3.32) | 16.13 ( 2.22) |

standard CNN with no invariance ($\mathcal{F}_\varnothing$) is typically very low except in Table 5 where sensitivity to all groups is required. We can also see the importance of $R(\mathcal{F})$ for tie-breaking in these experiments. As discussed in Section 4.2, $\widehat{\text{COMP}}(\mathcal{F}, \mathcal{D})$ is unable to distinguish between the different function classes because the training data contains a single example per equivalence class (see Figure 2c).

### A.5 CIFAR10 EXPERIMENTS WITH INFINITE/NONGROUP TRANSFORMATION SETS

In this section, we test our proposed method on out-of-distribution tasks on CIFAR10 images (Krizhevsky et al., 2009) where the equivalence relations are provided as infinite sets of transformations that may not form a group. We used (a) arbitrary rotation transformations over an image (denoted $\mathcal{T}_{\text{rot}}$), and (b) shifting the hue of an image (denoted $\mathcal{T}_{\text{col}}$). Note that for a bounded image, arbitrary rotation is not a group due to cropping. Further, transformations from the respective sets commute with each other, and hence, the lumpability condition is satisfied (Definition 2) for the corresponding equivalence relations.

We tested our method on 2 classification tasks: (i) invariant to both sets of transformations (arbitrary rotations and hue shifts), and (ii) invariant to arbitrary rotations, but sensitive to hue shifts. As before, we use the VGG architecture (Simonyan & Zisserman, 2014) for image classification and construct a collection of function classes $\mathscr{F} := \{\mathcal{F}_{\mathbb{S}} : \mathbb{S} \subseteq \{\text{rot}, \text{col}\}\}$ corresponding to the various invariant representations. We use data augmentation to construct these invariant representations (this is possible since the lumpability condition holds). For example, $\mathcal{F}_{\{\text{rot,col}\}}$ refers to CNNs that were trained by

Table 6: Results for different function classes on the CIFAR10 classification task with two sets of transformations (transformations that do not form groups) on images: *arbitrary rotations* (with cropping due to rotation) and *arbitrary hue shifts*. The task is invariant to both sets of transformations ($\bar{\mathbb{D}}$) and sensitive to none ($\mathbb{D}$). $R(\mathcal{F})$, $\widehat{\text{COMP}}(\mathcal{F}, \mathcal{D})$ and $S(\mathcal{F}, \mathcal{D})$ are as discussed in Section 4.2. **Bold** values indicate the function class chosen by GES method with the proposed scoring criterion. Test accuracy is computed on the extrapolated dataset after shifting the distribution of $P(\{U_i\}_{i \in \bar{\mathbb{D}}})$. We see that the $S(\mathcal{F}, \mathcal{D})$ loss selects the correct model class in training.

| Model class | $R(\mathcal{F})$ | $+ \widehat{\text{COMP}}(\mathcal{F}, \mathcal{D})$ | $+ \text{NLL}(\mathcal{F}, \mathcal{D})$ | $= S(\mathcal{F}, \mathcal{D})$ | Train Acc | Test Acc |
|---|---|---|---|---|---|---|
| $\mathcal{F}_{\{\}}$ | 3 | 27725.875 | 17496.615 | 45225.490 | 85.60 | 21.48 |
| $\mathcal{F}_{\{\text{col}\}}$ | 1 | 27716.947 | 22715.956 | 50433.903 | 81.28 | 21.85 |
| $\mathcal{F}_{\{\text{rot}\}}$ | 1 | -60894.145 | 20365.793 | -40527.352 | 82.65 | 45.12 |
| $\mathcal{F}_{\{\text{rot,col}\}}$ | **0** | **-66262.157** | **23538.768** | **-42723.390** | **79.99** | **69.35** |

Table 7: Results for different function classes on the CIFAR10 classification task with two sets of transformations (transformations that do not form groups) on images: *arbitrary angle rotations* (with cropping due to rotation) and *arbitrary hue shifts*. The task is invariant to arbitrary rotations of the image ($\bar{\mathbb{D}}$) but sensitive to color ($\mathbb{D}$). $R(\mathcal{F})$, $\widehat{\text{COMP}}(\mathcal{F}, \mathcal{D})$ and $S(\mathcal{F}, \mathcal{D})$ are as discussed in Section 4.2. **Bold** values indicate the function class chosen by GES method with the proposed scoring criterion. Test accuracy is computed on the extrapolated dataset after shifting the distribution of $P(\{U_i\}_{i \in \bar{\mathbb{D}}})$. We see that the $S(\mathcal{F}, \mathcal{D})$ loss selects the correct model class in training.

| Model class | $R(\mathcal{F})$ | $+ \widehat{\text{COMP}}(\mathcal{F}, \mathcal{D})$ | $+ \text{NLL}(\mathcal{F}, \mathcal{D})$ | $= S(\mathcal{F}, \mathcal{D})$ | Train Acc | Test Acc |
|---|---|---|---|---|---|---|
| $\mathcal{F}_{\{\}}$ | 3 | 27724.256 | 42166.993 | 69894.250 | 64.37 | 17.16 |
| $\mathcal{F}_{\{\text{col}\}}$ | 1 | 27715.023 | 49744.680 | 77460.703 | 42.69 | 10.91 |
| $\mathcal{F}_{\{\text{rot}\}}$ | **1** | **-91370.533** | **46218.086** | **-45151.447** | **61.77** | **52.60** |
| $\mathcal{F}_{\{\text{rot,col}\}}$ | 0 | -92009.184 | 50246.908 | -41762.276 | 41.45 | 35.56 |

augmenting both arbitrarily rotated images and hue-shifted images. Once again, $\mathcal{F}_\varnothing$ is the space of functions with no invariance (standard CNN with no data augmentations).

**Results.** We show in Tables 6 and 7 that our method is able to find the correct invariance and achieves the best OOD test accuracy whereas the standard CNN with no invariance has poor OOD performance.

A.6 MORE ON LUMPABILITY (DEFINITION 2)

We show that the lumpability condition of Definition 2 is equivalent to the normal subgroup condition of Mouli & Ribeiro (2021, Theorem 2) when the given equivalence relations are obtained from transformation groups. However, unlike the normal subgroup condition, the lumpability condition applies in the general case when the equivalence relations are not necessarily obtained via transformation groups.

**Proposition 1.** *Let $\sim_{G_1}$ and $\sim_{G_2}$ be two equivalence relations on the input space $\mathcal{X}$ obtained as orbits under transformation groups $G_1$ and $G_2$ respectively, i.e., for $i = 1, 2$, $\boldsymbol{x} \sim_{G_i} \boldsymbol{x}'$ iff there exists $t^{(i)} \in G_i$ with $\boldsymbol{x}' = t^{(i)} \circ \boldsymbol{x}$. Then, $\sim_{G_1}$ is lumpable with respect to the transformations $G_2$ (Definition 2) if and only if $G_1$ is a normal subgroup of $G_1 \vee G_2$, where $\vee$ is the join operator.*

*Proof.* First, given $\sim_{G_1}$ is lumpable with respect to $G_2$, we wish to prove that $G_1$ is a normal subgroup of $G_1 \vee G_2$. By definition of the join operator on transformation groups, $G_1$ is a subgroup of $G_1 \vee G_2$.

Next, consider an equivalence class $[\boldsymbol{x}]_{G_1} \in \mathcal{X}/\sim_{G_1}$. Then, by the lumpability of $\sim_{G_1}$ with respect to $G_2$, we have that for all $t^{(2)} \in G_2$, there exists $[\boldsymbol{x}']_{G_1}$ with $\boldsymbol{x}^* \in [\boldsymbol{x}]_{G_1} \implies t^{(2)} \circ \boldsymbol{x}^* \in [\boldsymbol{x}']_{G_1}$. In other words, each $t^{(2)}$ maps all points in one equivalence class $[\boldsymbol{x}]_{G_1}$ to another equivalence class

$[\boldsymbol{x}']_{G_1}$. Specifically, $t^{(2)}$ maps $\boldsymbol{x} \in [\boldsymbol{x}]_{G_1}$ to $t^{(2)} \circ \boldsymbol{x} \in [\boldsymbol{x}']_{G_1}$. Thus, we can set $\boldsymbol{x}' = t^{(2)} \circ \boldsymbol{x}$ without loss of generality.

Then, for all $t^{(2)} \in G_2$, we have from the lumpability condition that

$$\boldsymbol{x}^* \in [\boldsymbol{x}]_{G_1} \implies t^{(2)} \circ \boldsymbol{x}^* \in [t^{(2)} \circ \boldsymbol{x}]_{G_1} . \tag{9}$$

Recall from the definition of the equivalence class derived from a transformation group (i.e., the orbit) that $\boldsymbol{x}^* \in [\boldsymbol{x}]_{G_1}$ means that there exists a transformation $t^{(1)} \in G_1$ that maps $\boldsymbol{x}$ to $\boldsymbol{x}^*$, i.e., $\boldsymbol{x}^* = t^{(1)} \circ \boldsymbol{x}$. Similarly, $t^{(2)} \circ \boldsymbol{x}^* \in [t^{(2)} \circ \boldsymbol{x}]_{G_1}$ means that there exists another transformation $\tilde{t}^{(1)}$ such that $t^{(2)} \circ \boldsymbol{x}^* = \tilde{t}^{(1)} \circ t^{(2)} \circ \boldsymbol{x}$.

Equation (9) then becomes

$$\exists t^{(1)} \in G_1 \text{ s.t. } \boldsymbol{x}^* = t^{(1)} \circ \boldsymbol{x} \implies \exists \tilde{t}^{(1)} \in G_1 \text{ s.t. } t^{(2)} \circ \boldsymbol{x}^* = \tilde{t}^{(1)} \circ t^{(2)} \circ \boldsymbol{x} , \tag{10}$$

for all $t^{(2)} \in G_2$.

Since Equation (10) holds for all $\boldsymbol{x}^* \in [\boldsymbol{x}]_{G_1}$ and for all $\boldsymbol{x} \in \mathcal{X}$, we have $\forall t^{(2)} \in G_2, \forall t^{(1)} \in G_1, \exists \tilde{t}^{(1)} \in G_1$ such that,

$$t^{(2)} \circ t^{(1)} = \tilde{t}^{(1)} \circ t^{(2)} ,$$

which implies that $G_1$ is a normal subgroup of $G_1 \vee G_2$. The converse can be proved trivially by reversing the steps of the above proof.

$\square$

