# OpenReview forum: "Asymmetry Learning for Counterfactually-invariant Classification in OOD Tasks"
_ICLR.cc/2022/Conference — ICLR 2022 Oral_

### Official Review · Reviewer_XTXH · 2021-10-31

**Correctness:** 4
**Technical Novelty And Significance:** 3
**Empirical Novelty And Significance:** 3
**Recommendation:** 8
**Confidence:** 2

**Main Review:**

Strengths:
This paper does a good job of describing an interesting and important problem, invariance to OOD symmetry transformations, and motivates why existing approaches may not work well. The approach novel as far as I'm aware, and clearly describes relevant concepts such as defining what we would want our OOD-transformation invariant representations to satisfy and how learning them is linked to causal structure discovery.

Weaknesses:
- The method appears to identify which of some set of transformations one should be invariant to (and learn correspondingly invariant representations). But we must first have prepared a collection of all possible symmetry transformations, which seems like a limitation of the method. Additionally, is there a trade-off where considering more symmetry transformations makes the method slower?
-  The experiments and comparisons are relatively minimal and are only in a toy environment. From the description of the method I assume there are computational difficulties in practically implementing this method for more "real-world" problems. The papers would be strengthened by either having larger more realistic experimental comparisons, or barring that an explanation of the practical difficulties of applying the method to larger problems and future steps that could resolve them.

**Summary Of The Paper:**

This paper considers a class of out of distribution (OOD) problems where at test time there may be new symmetry transformations of the input X (i.e., they don't change the label Y). The authors explain why standard invariances learned by data augmentation may not be OOD invariant. Next, the paper presents a method for learning OOD-invariant representations through causal structure discovery. This hinges on the concept of being counterfactually invariant to the symmetry transformations that could appear at test time but don't affect Y. Next is an algorithm for discovering the structure of a causal DAG which largely revolves around deciding whether or not there is an edge U_i -> Y; this existence question corresponds to whether or not Y is invariant to the transformation U_i (IIUC).

The paper test this approach on tasks in a simple simulated physics environment.

**Summary Of The Review:**

Interesting ideas with relatively toy empirical results.

---

> ### Author Response · Authors · 2021-11-16
> **Official Response to Reviewer XTXH (Part 2/2)**
>
> **Q4:** The experiments and comparisons are relatively minimal and are only in a toy environment. From the description of the method I assume there are computational difficulties in practically implementing this method for more "real-world" problems. The papers would be strengthened by either having larger more realistic experimental comparisons, or barring that an explanation of the practical difficulties of applying the method to larger problems and future steps that could resolve them.
>
> **A4:** We have not experienced computational difficulties in implementing the method. The computation of COMP in Equation (7) can be approximated by sampling random labels and fitting the model to these random labels (amounts to training the model on new datasets). In our experiments, we estimate the complexity by Monte Carlo sampling 5 different random labelings. Additionally, as discussed above, Greedy Equivalence Search does not need to evaluate all the 2^m possible invariant representations (the greedy complexity is O(m)).
>
> We have updated our results section with additional experiments on images. Appendix A.4 details the experiments on MNIST-{3,4} images where we are given finite transformation groups. Appendix A.5 details the experiments on CIFAR10 images where we are given infinite transformation sets that may not form a group. Please see the details below.
>
>
> **MNIST-{3,4} with finite transformation groups.** We follow the out-of-distribution experiments of [2] where the equivalence relations are provided as transformation groups (e.g., $90^\circ$ rotations) over images. We use the Colored-MNIST-{3,4} dataset [2] that only contains digits 3 and 4. This was done only to avoid any confounding factors while testing if the proposed method can learn the correct invariances, not for any practical considerations (e.g., rotated 6 is a 9 and would interfere with some experiments, etc.).
>
> We consider equivalence relations obtained from 3 different transformation groups: rotations by $90^\circ$, vertically flipping the image, and permuting the RGB color channels of the image.
> We test our method on 4 classification tasks where each task represents the case where the target $Y$ has different invariances, i.e., invariant to all three groups, to two, to one, invariant to none (and sensitive to the remaining groups).
>
> We show in Tables 2&3 in the paper that our method is able to find the correct invariances and the best OOD accuracy. In Tables 4&5, the method is excessively invariant (to vertical flip) but still achieves within 1% of the best OOD accuracy. The OOD test accuracy of standard CNN with no invariance ($\mathcal{F}_\emptyset$) is typically very low except in Table 5 where sensitivity to all groups is required.
>
> **CIFAR10 experiments with infinite/nongroup transformation sets.** We also tested our method when a given set of transformations over the images does not form a group. We used (a) arbitrary rotation transformations over an image and (b) shifting the hue of an image. Note that for a bounded image, arbitrary rotation is not a group due to image cropping.
> We tested our method on 2 classification tasks: (i) invariant to both sets of transformations, and (ii) invariant to arbitrary rotations, but sensitive to hue shifts.
>
> We show in Tables 6&7 in the paper that our method is able to find the correct invariance and achieves the best OOD accuracy whereas the standard CNN with no invariance has poor OOD performance.
>
> **References:**
>
> [1] Martin Arjovsky, Léon Bottou, Ishaan Gulrajani, and David Lopez-Paz. Invariant risk minimization.arXiv preprint arXiv:1907.02893, 2019.
>
> [2]  S Chandra Mouli and Bruno Ribeiro. Neural network extrapolations with G-invariances from a single environment. In International Conference on Learning Representations, 2021.

---

> > ### Author Response · Authors · 2021-11-22
> > **Other questions/comments?**
> >
> > We believe we have addressed all your concerns. Please let us know if you have other questions or comments.

---

> ### Author Response · Authors · 2021-11-16
> **Official Response to Reviewer XTXH (Part 1/2)**
>
> We thank the reviewer for the positive comments and valuable feedback.
>
> **Q1:** The method appears to identify which of some set of transformations one should be invariant to (and learn correspondingly invariant representations).
>
> **A1:** A minor clarification: Given $m$ sets of symmetry transformations, the method finds which of these sets of transformations it should be invariant to. The method does not find which of the individual transformations within a particular set one needs to be invariant to.
>
> **Q2:** But we must first have prepared a collection of all possible symmetry transformations, which seems like a limitation of the method.
>
> **A2:** Out-of-distribution prediction without test distribution examples always requires additional assumptions or domain information: It is, unfortunately, a requirement of all causal queries. Without assumptions, the labels can arbitrarily change for out-of-distribution inputs. For instance, domain adaptation methods assume the availability of test distribution of the covariates $X$ and that $P(Y|X)$ remains the same, Invariant risk minimization (IRM) [1] and related methods assume the availability of data collected under multiple distinct environments observed in test. In this work, we use possible input symmetries as the domain knowledge. While the practitioner has to specify a collection of possible symmetries, we allow the flexibility of them being incorrect for the task at hand, since our method will choose the symmetries that are independent of the target $Y$ in training.
>
> **Q3:** Additionally, is there a trade-off where considering more symmetry transformations makes the method slower?
>
> **A3:** If we are given $m$ possible sets of symmetry transformations, the computational cost is O(m). In our causal search of symmetries, we use Greedy Equivalence Search (GES) described at the end of Section 4.2 that tests O(m) invariant representations instead of the prohibitive O(2^m) possible invariant representations. In all our experiments, this is enough to find the right invariances that achieve good OOD accuracy.

---

### Official Review · Reviewer_RsMV · 2021-10-31

**Correctness:** 4
**Technical Novelty And Significance:** 3
**Empirical Novelty And Significance:** 2
**Recommendation:** 8
**Confidence:** 3

**Main Review:**

**Update**: After reading the rebuttal and the other reviews, I have decided to bump up my original score to an 8. Thank you to the authors for addressing our concerns in great detail.

**Strengths**:
- The methodology has a strong theoretical basis, and is a result of combining insight from different fields. The idea behind this approach, is in my opinion, quite elegant.
- The paper is well-written and the work is well-placed in the literature, the illustrations are well-thought-out and facilitate understanding the method.

**Weaknesses**:
- The experimental section consists of only two simulated toy examples. While these showcase how the method is supposed to work, it remains unclear how well the method would work in a real-world, uncontrolled environment. How often are the assumptions made about the data generating process expected to hold 'in the wild'?

Other comments:
- The figures and table appear out of place. Table 1 is referenced in the Results section, but is shown two sections before without context. Figure 1 appears a bit too early in the paper (first mentioned in "Illustrative SCM example"), while Figure 2 appears much later (first mentioned before Figure 1, after that mentioned before Theorem 1).
- The caption text for Figure 2a(iii) is clipped out. The subfigure is also way too small to be readable without zooming in.
- Reference to "Accounting for unobserved confounding in domain generalization" appears to be duplicated.
- page 2, Layer 1 - "causal" misspelled as "casual"
- page 3, Symmetry transformations - Sentence starting with "Although" appears incomplete. I would suggest conjoining this sentence and the one before: "... equivalence classes, although ...".
- page 8, Causal structure discovery, third row from the bottom: "between distribution" -> "between **a** distribution"
- page 9, second row: add comma before "under the assumption"

**Summary Of The Paper:**

The authors propose an approach for constructing classifiers that achieve out-of-distribution (OOD) generalization using a new learning paradigm they call _asymmetry learning_. They consider OOD tasks where the test input is obtained from the training input by applying a sequence of (random) input transformations. To obtain an invariant OOD classifier that generalizes well to both in-distribution and out-of-distribution samples, the authors introduce the concept of _counterfactual invariant representations over symmetric transformations_. They show how learning the invariant representations can be cast as a causal structure discovery task and propose a score-based (GES-based) algorithm for finding the causal directed acyclic graph (DAG) that best describes the invariances.

**Summary Of The Review:**

The authors propose an interesting idea for constructing OOD classifiers starting from counterfactual invariance for symmetric transformation, an idea that is well-formulated in this paper. Despite the lack of real-world validation, I think the paper has a strong theoretical basis and is a decent contribution to the literature.

---

> ### Author Response · Authors · 2021-11-16
> **Official Response to Reviewer RsMV**
>
> We thank the reviewer for the positive comments and valuable feedback.
>
> **Q1:** The experimental section consists of only two simulated toy examples. While these showcase how the method is supposed to work, it remains unclear how well the method would work in a real-world, uncontrolled environment.
>
> **A1:** Thank you for the feedback. We have updated our results section with additional experiments on images. Appendix A.4 details the experiments on MNIST-{3,4} images where we are given finite transformation groups. Appendix A.5 details the experiments on CIFAR10 images where we are given infinite transformation sets that may not form a group. Please see the details below.
>
>
> **MNIST-{3,4} with finite transformation groups.** We follow the out-of-distribution experiments of [1] where the equivalence relations are provided as transformation groups (e.g., $90^\circ$ rotations) over images. We use the Colored-MNIST-{3,4} dataset [1] that only contains digits 3 and 4. This was done only to avoid any confounding factors while testing if the proposed method can learn the correct invariances, not for any practical considerations (e.g., rotated 6 is a 9 and would interfere with some experiments, etc.).
>
> We consider equivalence relations obtained from 3 different transformation groups: rotations by $90^\circ$, vertically flipping the image, and permuting the RGB color channels of the image.
> We test our method on 4 classification tasks where each task represents the case where the target $Y$ has different invariances, i.e., invariant to all three groups, to two, to one, invariant to none (and sensitive to the remaining groups).
>
> We show in Tables 2&3 in the paper that our method is able to find the correct invariances and the best OOD accuracy. In Tables 4&5, the method is excessively invariant (to vertical flip) but still achieves within 1% of the best OOD accuracy. The OOD test accuracy of standard CNN with no invariance ($\mathcal{F}_\emptyset$) is typically very low except in Table 5 where sensitivity to all groups is required.
>
> **CIFAR10 experiments with infinite/nongroup transformation sets.** We also tested our method when a given set of transformations over the images does not form a group. We used (a) arbitrary rotation transformations over an image and (b) shifting the hue of an image. Note that for a bounded image, arbitrary rotation is not a group due to image cropping.
> We tested our method on 2 classification tasks: (i) invariant to both sets of transformations, and (ii) invariant to arbitrary rotations, but sensitive to hue shifts.
>
> We show in Tables 6&7 in the paper that our method is able to find the correct invariance and achieves the best OOD accuracy whereas the standard CNN with no invariance has poor OOD performance.
>
>
> **Q2:** How often are the assumptions made about the data generating process expected to hold 'in the wild'?
>
> **A2:** Thanks for the question. It really depends on the type of task. We expect physics tasks to have a number of symmetries (e.g., conservation of energy, conservation of momentum, conservation of angular momentum, etc.). Image tasks tend to be symmetric to some types of color changes (brightness, hue, small color balance differences), image scaling symmetries, limited angle rotation symmetries, and symmetries to small occlusions (modern self-supervised learning methods seem to rely on these small occlusions). Time series may be time stationary (which is a time shift symmetry) and may be ergodic (which is a time average symmetry).
>
> **Other comments:**
>
> **Q3:** The figures and table appear out of place. Table 1 is referenced in the Results section, but is shown two sections before without context. Figure 1 appears a bit too early in the paper (first mentioned in "Illustrative SCM example"), while Figure 2 appears much later (first mentioned before Figure 1, after that mentioned before Theorem 1).
>
> **A3:** We have moved Table 1 and Figure 2 and updated their references.
>
> **Q4:** The caption text for Figure 2a(iii) is clipped out. The subfigure is also way too small to be readable without zooming in.
>
> **A4:** We have fixed the caption and increased the font sizes of all the figures.
>
> **Typos:** We have fixed these typos (and other small typos) in the paper, thanks.
>
> **References:**
>
> [1] S Chandra Mouli and Bruno Ribeiro. Neural network extrapolations with G-invariances from a single environment. In International Conference on Learning Representations, 2021.

---

> > ### Author Response · Authors · 2021-11-22
> > **Other questions/comments?**
> >
> > We believe we have addressed all your concerns. Please let us know if you have other questions or comments.

---

> > > ### Comment · Reviewer_RsMV · 2021-11-22
> > > **Acknowledgment**
> > >
> > > Thank you very much for addressing the few concerns I had.

---

### Official Review · Reviewer_GzS3 · 2021-11-04

**Correctness:** 3
**Technical Novelty And Significance:** 3
**Empirical Novelty And Significance:** 3
**Recommendation:** 8
**Confidence:** 4

**Main Review:**

- The paper is well motivated, and examines an important problem of classifiers failing out-of-domain.
- The paper's contributions could be significant to what is an emerging area of research. Other than [1, 2] and [3], I have not seen any other paper that offers a theoretical framing for the relationship between counterfactual invariance and OOD generalization and provides a way to achieve the same.
- The paper flows smoothly and is easy to understand.
- I wish the paper offered a greater discussion of prior work on counterfactual invariance as it relates to out-of-domain generalization. I hope the authors can address that in a future version. Additionally, while the authors have offered some discussion of [2] and [3], per my reading, [1] seems to offer a contradictory view (from this paper) of whether learning an invariant OOD classifier is solvable via interventional data augmentation or not. It would be great if the authors could share how one relates to the other. I'm not sure I found a convincing argument laid in this paper other than the example in Figure 1(c).
- The experiments in the paper are on simulated tasks with limited number of variables. It is unclear how this method might work in high dimensional spaces, such as text. Though it's not necessarily a concern, I think it would be useful if the authors could comment on the same.

[1] Kaushik, D., Setlur, A., Hovy, E. H., & Lipton, Z. C. Explaining the Efficacy of Counterfactually Augmented Data. ICLR 2021.
[2] Veitch et al. (2021)
[3] Wang and Jordan (2021)

Typos and presentation:
- Introduction Line 2: "the task is requires" -> "the task requires"
- Break the second sentence of the first paragraph of introduction.
- Layer 3 description of Pearl's causal hierarchy: "X\dagger describe an hidden variable" -> "X\dagger describe a hidden variable"
- Your figures are blurry upon printing the paper. Please increase the font of the text in the figures.
- Section 4.1: "Our next results requires imposing" -> "Our next results require imposing"
- Section 4.1 under Definition 2: "Definition 2 holds for a equivalence relation" -> "Definition 2 holds for an equivalence relation"
- Related Work should either be Section 6 or be Section 2 (right after introduction) and not be in between the theory and empirical results.
- In justifying Assumption 1, can you provide a citation that is more recent considering the science in this area has evolved considerably since 1982?

**Summary Of The Paper:**

Update

I have read the author response and the updated version of this paper. I am delighted to see that the authors have incorporated my feedback and I believe this makes the paper stronger than its previous version. I have updated my scores and recommend that the paper be accepted.

------------------------------------------------------------------

This paper proposes Asymmetry Learning, a new learning paradigm to obtain counterfactually invariant classifiers. If the observed covariates are a result of some transformations applied to a hidden variable, and these transformations differ between training and test datasets, a classifier may not generalize to the test set. The authors argue that when these transformations are a result of a collection of equivalence relations, finding OOD-invariant classifier boils down to finding the simplest causal model that defines the causal relationships between the labels and these symmetry transformations. To this end, the authors propose a scoring criterion to identify the simplest DAG in a DAG search space that ensures that the label is invariant of all transformations while maximizing the likelihood of the observed training set. Using their proposed scoring function, the authors employ Greedy Equivalence Search to identify the DAG with the highest score. Experiments on simulated physics tasks suggest that the method works as intended.

**Summary Of The Review:**

Overall, I think the paper presents a good contribution but I do have some minor concerns that I would like the authors to address as I have mentioned above.

---

> ### Author Response · Authors · 2021-11-16
> **Official Response to Reviewer GzS3 (Part 2/2)**
>
> **Q2:** The experiments in the paper are on simulated tasks with limited number of variables. It is unclear how this method might work in high dimensional spaces, such as text. Though it's not necessarily a concern, I think it would be useful if the authors could comment on the same.
>
> **A2:** Thank you for the feedback. We have updated our results section with additional experiments on images. Appendix A.4 details the experiments on MNIST-{3,4} images where we are given finite transformation groups. Appendix A.5 details the experiments on CIFAR10 images where we are given infinite transformation sets that may not form a group. Please see the details below.
>
> **MNIST-{3,4} with finite transformation groups.** We follow the out-of-distribution experiments of [5] where the equivalence relations are provided as transformation groups (e.g., $90^\circ$ rotations) over images. We use the Colored-MNIST-{3,4} dataset [5] that only contains digits 3 and 4. This was done only to avoid any confounding factors while testing if the proposed method can learn the correct invariances, not for any practical considerations (e.g., rotated 6 is a 9 and would interfere with some experiments, etc.).
>
> We consider equivalence relations obtained from 3 different transformation groups: rotations by $90^\circ$, vertically flipping the image, and permuting the RGB color channels of the image.
> We test our method on 4 classification tasks where each task represents the case where the target $Y$ has different invariances, i.e., invariant to all three groups, to two, to one, invariant to none (and sensitive to the remaining groups).
>
> We show in Tables 2&3 in the paper that our method is able to find the correct invariances and the best OOD accuracy. In Tables 4&5, the method is excessively invariant (to vertical flip) but still achieves within 1% of the best OOD accuracy. The OOD test accuracy of standard CNN with no invariance ($\mathcal{F}_\emptyset$) is typically very low except in Table 5 where sensitivity to all groups is required.
>
> **CIFAR10 experiments with infinite/nongroup transformation sets.** We also tested our method when a given set of transformations over the images does not form a group. We used (a) arbitrary rotation transformations over an image and (b) shifting the hue of an image. Note that for a bounded image, arbitrary rotation is not a group due to image cropping.
> We tested our method on 2 classification tasks: (i) invariant to both sets of transformations, and (ii) invariant to arbitrary rotations, but sensitive to hue shifts.
>
> We show in Tables 6&7 in the paper that our method is able to find the correct invariance and achieves the best OOD accuracy whereas the standard CNN with no invariance has poor OOD performance.
>
>
> **Typos**: We have fixed these typos (and other small typos) in the paper, thanks.
>
> **Presentation:** We have fixed the sentence in the introduction, increased the fonts in all figures, and moved related work to Section 6.
>
> **Q3:** In justifying Assumption 1, can you provide a citation that is more recent considering the science in this area has evolved considerably since 1982?
>
> **A3:** We provide a more recent reference on human visual perception and symmetries (Westphal-Fitch et al. (2012)) to justify Assumption 1.
>
> **References:**
>
> [1] Kaushik, D., Setlur, A., Hovy, E. H., & Lipton, Z. C. Explaining the Efficacy of Counterfactually Augmented Data. ICLR 2021.
>
> [2] Victor Veitch, Alexander D’Amour, Steve Yadlowsky, and Jacob Eisenstein. Counterfactual invariance to spurious correlations: Why and how to pass stress tests. NeurIPS 2021
>
> [3] Yixin Wang and Michael I. Jordan. Desiderata for Representation Learning: A Causal Perspective. arXiv:2109.03795 [cs, stat], September 2021.
>
> [4] Kaushik, Divyansh, Eduard Hovy, and Zachary C. Lipton. "Learning the difference that makes a difference with counterfactually-augmented data." arXiv preprint arXiv:1909.12434 (2019).
>
> [5] S Chandra Mouli and Bruno Ribeiro. Neural network extrapolations with G-invariances from a single environment. In International Conference on Learning Representations, 2021.
>
> [6] Shorten, Connor, and Taghi M. Khoshgoftaar. "A survey on image data augmentation for deep learning." Journal of Big Data 6.1 (2019): 1-48.
>
> [7] Gesche Westphal-Fitch, Ludwig Huber, Juan Carlos Gomez, and W Tecumseh Fitch. Production and perception rules underlying visual patterns: effects of symmetry and hierarchy. Philosophical Transactions of the Royal Society B: Biological Sciences, 367(1598):2007–2022, 2012.

---

> > ### Author Response · Authors · 2021-11-22
> > **Other questions/comments?**
> >
> > We believe we have addressed all your concerns. Please let us know if you have other questions or comments.

---

> ### Author Response · Authors · 2021-11-16
> **Official Response to Reviewer GzS3 (Part 1/2)**
>
> We thank the reviewer for the positive comments and valuable feedback.
>
> **Q1:** I wish the paper offered a greater discussion of prior work on counterfactual invariance as it relates to out-of-domain generalization. I hope the authors can address that in a future version. Additionally, while the authors have offered some discussion of [2] and [3], per my reading, [1] seems to offer a contradictory view (from this paper) of whether learning an invariant OOD classifier is solvable via interventional data augmentation or not. It would be great if the authors could share how one relates to the other. I'm not sure I found a convincing argument laid in this paper other than the example in Figure 1(c).
>
> **A1:** We have added a more in-depth comparison of our work with the existing counterfactual methods in Appendix A.3.
> [1, 4] propose counterfactual data augmentation for text datasets where human annotators are asked to make minimal modifications to the input document so as to change its label (for example, by changing a few positive words to negative words) while keeping style, etc. fixed. This type of augmentation essentially asks the labelers to identify all the causal features in the document and make modifications to those features alone. This can be seen as obtaining new counterfactual examples by simulating the causal model and requires knowing the true function that describes how the features affect the labels. We consider the more realistic setting where we do not have access to such a collection of counterfactual examples.
>
> We will clarify why the results of [1] are not contradictory to our paper. In our paper, data augmentation refers to the traditional automated interventional data augmentation [6] under a mostly unknown data generation process, as opposed to the counterfactual data augmentation in [1] that either considers a fully-specified toy SCM or relies on humans-in-the-loop to generate counterfactual data. In Figure 1(c) we show that interventional data augmentation is not sufficient for the OOD task. However, if one had access to the fully-specified causal model, one could generate the counterfactual data shown in Figure 1(d) and learn an OOD classifier with the counterfactually augmented data (as done in [1]). But our work does not assume access to these counterfactual examples as [1] did. We will make this distinction clear in the paper.
> Additionally, we prove that a counterfactual invariant classifier can be constructed from interventional data augmentation alone if the lumpability condition (Definition 2) is satisfied. This is not the case in Figure 1(d). In the trivial case, if one considers a single set of symmetry transformations, lumpability holds and interventional data augmentation is always enough.
>
> Wang & Jordan [3] use counterfactual language to formally define and learn non-spurious, disentangled representations from a single environment. Our work is different in the following ways. In the structural causal model (SCM) of [3], the authors assume that there are no confounders between the observed $X$ and the label $Y$. However, in our SCM, we allow unobserved confounders $X^\dagger$ and $U_i, i\in \mathbb{D}$. The hidden transformation variables $U_i, i\in \mathbb{D}$ are confounders because they affect both the observed input $X$ and the labels $Y$. We leverage the fact that confounders are related to symmetries (and do not affect X arbitrarily) to resolve the issue of unobserved confounding. Wang & Jordan also require pinpointability of the cause of the observed $X$. In our setting, this is typically not possible since there are multiple paths of transformations from $X^\dagger$ to the same observed $X$. Thus, all the parents of $X$ may not be pinpointable, specifically the transformation variables $U_1, … U_m$.
>
>  Veitch et al. (2021) [2] define counterfactual invariant predictors $f(X)$ when $X$ has a single parent $Z$ and provide conditions such predictors must satisfy over the observed distribution. Note that Veitch et al. assume that part of the input $X$ ($X^\perp_Z$) is not causally influenced by the confounder Z. In our scenarios this is not generally true. For instance, under a color change, the entire image $X$ changes. Still, we show that the notion of a counterfactual invariant predictor exists. Hence, the definition of Veitch et al. in Lemma 3.1 of a counterfactually invariant predictor that requires segment of X to not causally depend on Z, a fundamental result of their work, unfortunately **does not apply** to our setting (since X may have no such segment).

---

### Decision · Program_Chairs · 2022-01-20

**Decision:**

Accept (Oral)

**Comment:**

This paper proposes asymmetry learning for learning counterfactual classifiers, i.e. classifiers which are invariant to certain symmetry transformations w.r.t. hidden variables that differ between the training and test sets.

The reviewers universally agreed that the proposed setting, and theoretical contribution, were interesting and novel. They also praised the writing quality, but had some quibbles about the quality of the experiments, and discussion of prior work. Neither of these concerns were considered significant enough to be a barrier to acceptance, but the authors should try to improve them, if possible.